# Computer-vision object tracking for monitoring bottlenose dolphin habitat use and kinematics

**Joaquin Gabaldon**[1]*, **Ding Zhang**[2], **Lisa Lauderdale**[3], **Lance Miller**[3],
**Matthew Johnson-Roberson**[1,4], **Kira Barton**[1,2], **K. Alex Shorter**[2]

**1** Robotics Institute, University of Michigan, Ann Arbor, MI, United States of America, **2** Department of Mechanical Engineering, University of Michigan, Ann Arbor, MI, United States of America, **3** Chicago Zoological Society, Brookfield Zoo, Brookfield, IL, United States of America, **4** Naval Architecture and Marine Engineering, University of Michigan, Ann Arbor, MI, United States of America

* gabaldon@umich.edu

## Abstract

This research presents a framework to enable computer-automated observation and monitoring of bottlenose dolphins (*Tursiops truncatus*) in a zoo environment. The resulting approach enables detailed persistent monitoring of the animals that is not possible using manual annotation methods. Fixed overhead cameras were used to opportunistically collect ∼100 hours of observations, recorded over multiple days, including time both during and outside of formal training sessions, to demonstrate the viability of the framework. Animal locations were estimated using convolutional neural network (CNN) object detectors and Kalman filter post-processing. The resulting animal tracks were used to quantify habitat use and animal kinematics. Additionally, Kolmogorov-Smirnov analyses of the swimming kinematics were used in high-level behavioral mode classification. The object detectors achieved a minimum Average Precision of 0.76, and the post-processed results yielded $1.24 \times 10^7$ estimated dolphin locations. Animal kinematic diversity was found to be lowest in the morning and peaked immediately before noon. Regions of the zoo habitat displaying the highest activity levels correlated to locations associated with animal care specialists, conspecifics, or enrichment. The work presented here demonstrates that CNN object detection is viable for large-scale marine mammal tracking, and results from the proposed framework will enable future research that will offer new insights into dolphin behavior, biomechanics, and how environmental context affects movement and activity.

## Introduction

Direct observation of dolphins at accredited facilities and in the wild has been key to developing an understanding of the behavior and biomechanics of these animals. How the dolphins behave in the presence of conspecifics, interact and engage with their environment, or are affected by changes to their environment are all questions of interest. Ideally, these observations are made without modifying animal behavior, and in a manner that facilitates a

**Data Availability Statement:** Data and code for this publication can be found at the following location: https://www.ebi.ac.uk/biostudies/studies/S-BSST793.

**Funding:** This study was funded by the Granger Foundation and the Chicago Zoological Society (CZS). Authors L. Lauderdale and L. Miller are members of CZS and contributed to the study design, data collection and analysis, and manuscript revision.

**Competing interests:** The authors have declared that no competing interests exist.

quantitative comparison between conditions in the environment. In zoo settings there is a strong emphasis on behavioral monitoring to inform welfare practices [1–3]. Bottlenose dolphins, the most common cetacean in zoos and aquariums, are generally regarded as a species that thrives under professional care, though data-driven studies of behavior and welfare have been limited [4, 5]. The ability to quantify animal motion and location, both in the environment and with respect to other animals, is therefore critical in understanding their behavior.

Biomechanics and behavioral studies depend on animal-based measurements that are considered reliable and repeatable for the species of interest [2, 6–8], but direct measurements of animals in the marine environment can be challenging. In zoo environments, animals tend to be monitored using external sensors, such as cameras and hydrophones, placed in the environment [9, 10]. These sensors can be combined into networks to observe a majority of the animals' environment with a relatively small number of devices. While it is possible to continuously record the animals' environmental use and social interactions, these videos must be heavily processed to convert them into useful information. This processing is often performed by a trained expert, who watches and scores behavioral or tracking information from the data [2, 11–13]. Examples of such studies include monitoring the effects of human presence on animal behaviors, analysis of dolphin activity cycles and sleep patterns, and the evaluation of social interactions with conspecifics. Unfortunately, hand-tracking is time consuming and can be inefficient when hundreds of hours of data have been collected from multiple sensors. Recent efforts have been made to automate this process for cameras, primarily through heuristically-crafted computer-vision techniques [14, 15]. However, these techniques were either limited in execution due to prohibitive costs (e.g. funds for the hardware/installation of an extended multi-camera array), or required manual tuning to account for changing environmental conditions (e.g. lighting shifts throughout the day).

To address these gaps, this work uses a neural network-based computer-automated framework to quantify the positional states of multiple animals simultaneously in a zoo environment, and employs the framework to investigate the dolphins' day-scale swimming kinematics. Neural networks have demonstrated flexibility and robustness in tracking biological systems from image and video data [16, 17]. To this end, a state-of-the-art neural network object detection technique, Faster R-CNN [18], was chosen as the backbone of the animal detection method for its prioritization of accuracy and precision regardless of object size or density in the image, as opposed to a faster single-shot detector [19]. The Faster R-CNN detector structure has demonstrated its capabilities in both land [20] and marine [21] applications, and is considered a reliable option for challenging tracking tasks.

In this study, camera data were used to monitor the behavior of a group of marine mammals both qualitatively and quantitatively in a zoo setting. Camera-based animal position data were used to quantify habitat usage, as well as where and how the group of animals moved throughout the day. The position data were decomposed into kinematic metrics, and used to discriminate between two general movement states—static and dynamic—using the velocity of the tracked animals. A general ethogram of the animals' behaviors monitored in this research is presented in Table 1. Joint differential entropy computations were calculated using animal speed and heading data to provide an understanding of the dolphins' kinematic diversity. Kolmogorov-Smirnov statistical analyses of the kinematic metrics were used to compare movement patterns and activity levels over time and between behavioral conditions. The proposed framework and results presented here demonstrate the viability of computer-vision inspired techniques for this challenging monitoring problem, and will enable future studies to gain new insights into dolphin behavior and biomechanics.

**Table 1. Behavior condition ethogram of dolphins under professional care.**

| Category | Behavior | Definition |
|---|---|---|
| ITS (In Training Session) | Animal Care Session | Time period in which animal care specialists work with the dolphins to learn new behaviors or practice known behaviors without public audience. |
| ITS | Public Presentation | Time period in which animal care specialists work with the dolphins in front of an audience to present educational information to the public. |
| OTS (Out of Training Session) | Static | Animal movement state with little to no active fluking at a rate of speed less than 0.5 ms$^{-1}$. |
| OTS | Dynamic | Animal movement state with active fluking at a rate of speed greater than 0.5 ms$^{-1}$. |

## Materials and methods

### Experimental environment

Seven bottlenose dolphins of ages 5, 5, 14, 16, 17, 33, and 36 years with lengths of 247 ± 17 cm were observed using a dual-camera system in the Seven Seas building of the Brookfield Zoo, Brookfield IL. The complete environment consists of a main indoor habitat with public viewing, two smaller habitats behind the main area, and a medical habitat between the two smaller habitats. The main habitat (Fig 1, top), which was the focus of the experiment, is 33.5 m across, 12.2 m wide, and 6.7 m deep. The habitats are connected through a series of gates. During formal training sessions in the main habitat, animal care specialists primarily engage with the animals on the island between the gates to the other areas. There are underwater observation windows for the viewing public on the far side of the main habitat from the island (not shown), and smaller windows looking into the offices of the animal care specialists on the island and next to the right gate (Fig 1, bottom). Recordings of the main habitat took place across multiple days (between Feb. 6 and March 27, 2018), for varying portions of each day, for a total of 99.5 hours over 20 recordings. Data collection began at the earliest at 07:41 and ended at the latest at 16:21. During the recorded hours, the dolphins participated in four formal training sessions according to a regular, well-defined schedule set by the animal care specialists (ACSs).

A formal training session consisted of time in which the ACSs work with the dolphins to learn new behaviors or practice known behaviors. At the beginning of each formal training session, the dolphins were asked to maintain positions directly in front of the ACS (formally known as "stationing"). The animal care specialists then presented discriminative stimuli or gestures that indicated which behaviors they requested each dolphin produce. When the animals were in a formal training session (abbreviated ITS), they experienced two formats of training during the data collection period: non-public animal care sessions and public presentations. Time outside of formal training sessions (abbreviated OTS) was defined as when the animals were not interacting with ACSs. During the OTS time periods, the ACSs would provide enrichment objects for the animals to interact with and select which parts of the habitat the animals could access using gates on either side of the main island. The time intervals for the OTS and ITS blocks are displayed in Table 2. The study protocol was approved by the University of Michigan Institutional Animal Care and Use Committee and the Brookfield Zoo.

### Experimental equipment

Two AlliedVision Prosilica GC1380C camera sensors with Thorlabs MVL5M23 lenses were separately mounted in Dotworkz D2 camera enclosures, which were attached to 80/20 T-slotted aluminum framing. On the frame, the cameras were spaced approximately 2m apart. The

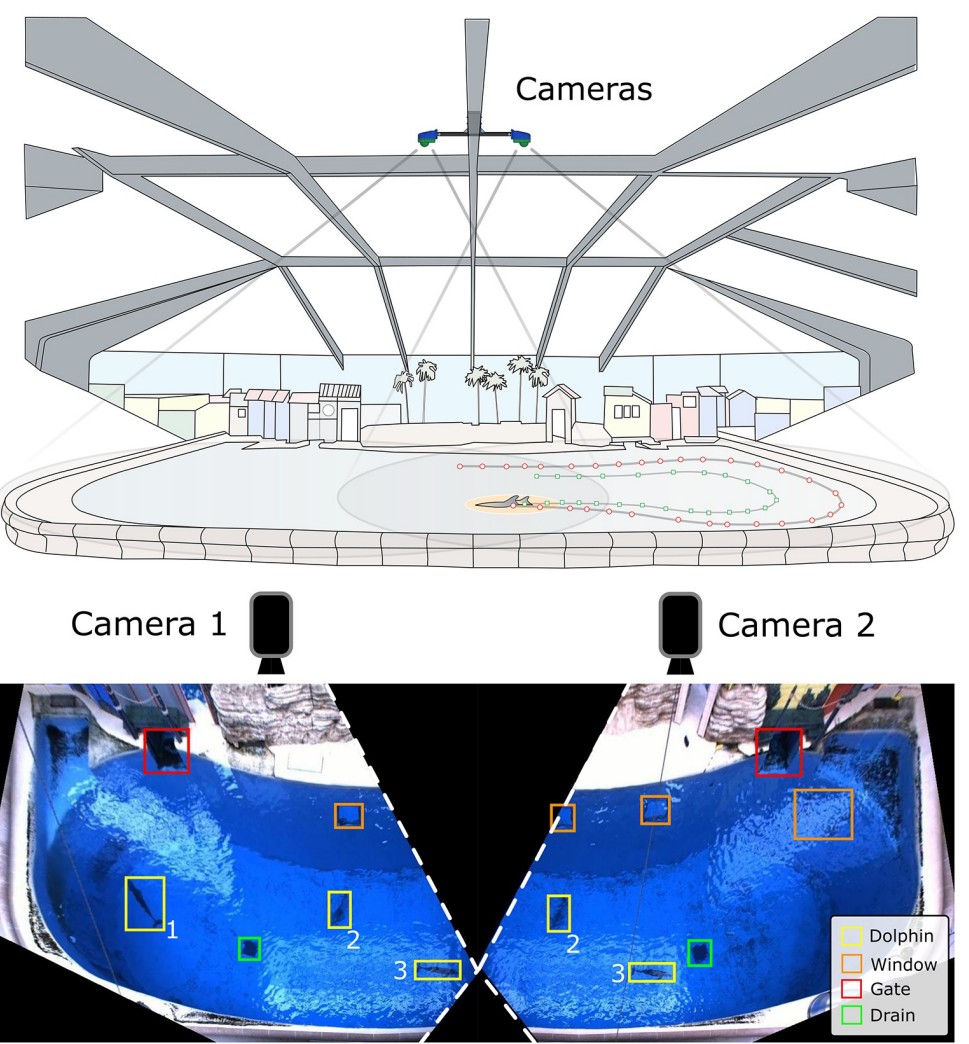

**Fig 1. Diagram of the experimental setup.** TOP: Illustration of the main habitat, with camera placements (blue enclosures) and fields of view (gray cones). BOTTOM: Top-down individual camera views, with objects in the habitat marked. Yellow { Dolphin bounding boxes, Green—Drains, Red—Gates between regions, Orange—Underwater windows (3 total). Correlated dolphin bounding boxes are indicated by number.

frame was mounted to a support beam directly above the main habitat, with the cameras angled to give full coverage of the area when combined. Fig 1, top, illustrates the habitat, camera placement, and field of view coverage. For data collection, the cameras were connected through the Gigabit Ethernet protocol to a central computer with an Intel i7–7700K CPU. Recordings were executed using the MATLAB Image Acquisition Toolbox, in the RGB24 color format at a frame rate of 20Hz. Each camera was connected to a separate Ethernet port on an internal Intel PRO/1000 Pt PCIe card. A separate computer system was used for detection inference, and was outfitted with an Intel i7–8700K processor clocked to 4.8 GHz and a Nvidia Titan V graphics processing unit in Tesla Compute Cluster mode.

## Dolphin detection

Approximately 99.5 hours of data from two cameras were collected for this work, resulting in ∼ 14 million individual frames of data. To extract spatial information about habitat use and

**Table 2. Block time intervals.**

| | Time Interval | |
|---|---|---|
| **Block** | **OTS** | **ITS** |
| 1 | 08:00–09:30 | 09:30–10:00 |
| 2 | 10:00–11:30 | 11:30–12:00 |
| 3 | 12:00–13:00 | 13:00–13:30 |
| 4 | 13:30–14:30 | 14:30–15:00 |
| 5 | 15:00–16:00 | N/A |
| | Dynamics Metrics (means) | |
| | Speed (ms$^{-1}$) | |
| 1 | 1.30 | 1.39 |
| 2 | 1.57 | 1.45 |
| 3 | 1.45 | 1.44 |
| 4 | 1.41 | 1.39 |
| 5 | 1.43 | N/A |
| | Yaw Rate (degs$^{-1}$) | |
| 1 | 0.32 | -0.68 |
| 2 | -3.61 | 1.28 |
| 3 | -0.18 | 3.26 |
| 4 | 0.05 | 2.53 |
| 5 | 1.99 | N/A |

The ITS blocks (1 and 3) are animal care sessions, and the OTS blocks (2 and 4) are public presentations. The corresponding mean speed and yaw rate dynamics metrics are also reported, with yaw rate converted to units of (degs$^{-1}$) for readability.

swimming kinematics, we first needed to identify animals in the frames. These detections were filtered and associated with short trajectories (tracklets) from individual animals. Kinematic data in the form of position, velocity, and yaw (heading in the *x-y* plane), from the tracklets were then used to parameterize and form probability distributions for each time block that were used to identify tendencies in animal motion during in training (ITS) and out of training session (OTS) swimming.

**Neural network methods.** The first step in the analysis process was dolphin detection from the captured video frames using Faster R-CNN, a machine-learning object detection method [18]. The method consisted of two primary modules: a Region Proposal Network (RPN), and a Fast R-CNN detector network. The RPN identified regions in an image that may enclose objects of interest, and presented these to the Fast R-CNN detector to verify which regions did in fact contain objects the detector sought to identify. These two modules when combined form one large network capable of returning a bounding box tightly enclosing an object's location within an image. For a more complete explanation of the method please refer to [18].

All modules used in the implementation were present in the MATLAB Deep Learning Toolbox excepting the Parametric Rectified Linear Unit (PReLU) activation function, which was defined with a custom neural network layer per directions in the MATLAB online documentation [22, 23]. The convolutional neural network (CNN) structure used in the Faster R-CNN framework is as follows. For the input layer, the size was chosen to be similar to the smallest bounding boxes in the set of manually scored dolphin profiles, in the format of (*l,l,*3), where *l* is 2× the side length of the smallest bounding box major axis. The input layer had a

third dimension of 3 as input images were in the RGB colorspace. The feature extraction layers had the following structure: four sets of 2D $3 \times 3$ convolution layers, each followed by batch normalization, PReLU activation, and $2 \times 2$ max pooling (stride 2) layers, in that order. The four convolution layers had, in order: 64, 96, 128, and 128 filters. Each convolution was performed with one layer of zero padding along the edges of the inputs to avoid discounting the corners/edges. The classification layers used the extracted features from the previous layers to identify an image region as either a dolphin or the background. They consisted of: 1) A fully connected layer, length 512, to extract features from the final convolution layer, followed by a PReLU activation; 2) A fully connected layer, length 2, to determine non-scaled classification weights; 3) A softmax function layer to convert these weights into the final probabilities of the image region's classification. The highest probability from the softmax layer corresponded to the most likely classification for the region, and the magnitude of this probability indicated the confidence of the classification.

**Training the network.** Ground truth data were scored by a trained observer who manually defined bounding boxes that identified the locations of the dolphins in the training/testing frames (Fig 1, bottom, yellow boxes). These ground truth data were selected over a range of lighting conditions and dolphin locations to ensure robustness of the detection network. For each camera, 100 frames were extracted from each of 11 separate recordings, with evenly spaced time intervals between frames. The recordings were collected in May 2017, and February, March, and August 2018. Over 940 frames from each of the left and right cameras were found to contain usable dolphin locations, i.e. human-detectable dolphin profiles. Each usable dolphin location in the selected frames was manually given a bounding box tightly enclosing the visible profile. The detector for the left camera was trained on 1564 profiles and tested on 662, and the detector for the right camera was trained on 1482 profiles and tested on 662. The dolphin detectors were trained using the MATLAB implementation of Faster R-CNN, employing the previously-defined CNN structure as the classification method.

**Detection processing.** Detections were performed over all 99.5 hours of recorded data from both cameras, at 10Hz intervals (total of $7.16 \times 10^6$ frames), using a 95% minimum confidence threshold to ensure accuracy. The fields of view of the two cameras overlap for a portion of the habitat, resulting in some dolphins being detected simultaneously by both cameras. This yielded multiple sets of conflicting detection bounding boxes spanning the two fields of view, which necessitated associating the most likely left/right box pairs. Before conflict identification was performed, the detection boxes were first transformed into a common plane of reference termed the world frame. Using known world point coordinates, homographies from each camera to the world frame were generated using the normalized Direct Linear Transform method [24]. These homographies were used to convert the vertices of the bounding boxes to the world frame using a perspective transformation. Intersecting boxes were identified by evaluating polygonal intersections, and Intersection over Union (IoU) metrics were computed for intersecting boxes to measure how well they matched. Associations were identified between pairs of left/right intersecting boxes with the highest mutual IoU values.

Associated boxes' world frame centroid locations were meshed using a weighted mean. First, the boundaries of each camera's field of view were projected into the world frame, allowing us to obtain the line in the world frame $y$-direction defining the center of the overlap region, denoted $l_s = x_{mid}$ (Fig 2, top, red lines). $x_{mid}$ is the $x$-coordinate in the world frame midway between the physical placement of the cameras. For each detection ($u$), the distance ($d_b$) in the $x$-direction from $u$ to the nearest projected camera boundary line ($b_n$) was then determined. Next, the distance ($d_l$) in the $x$-direction from line $l_s$ through $u$ to $b_n$ was found. Finally, the weight for the camera corresponding to $b_n$ was calculated as $w_n = d_b/2d_l$, with the weight

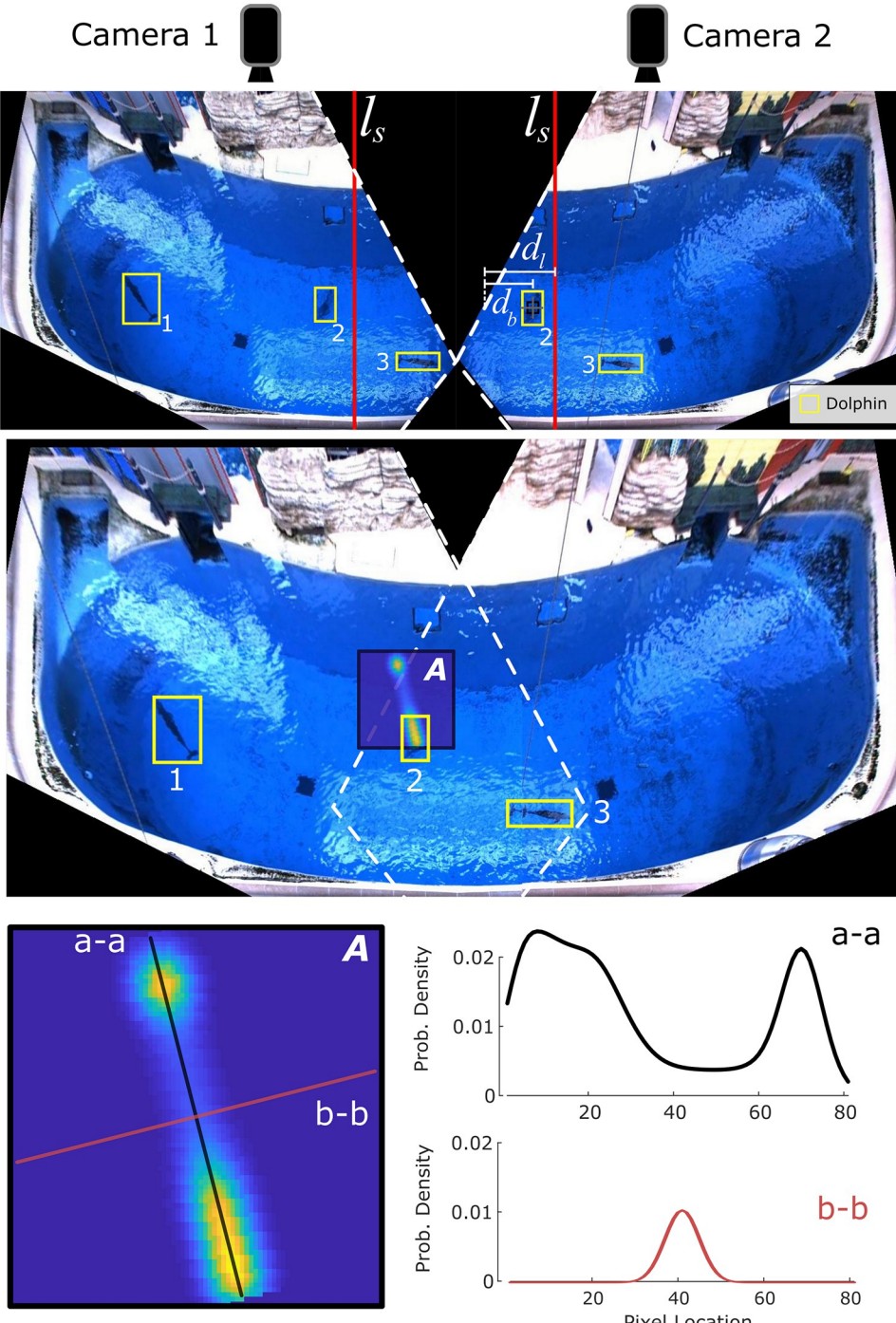

**Fig 2. Combined figure demonstrating camera overlap, bounding box meshing, and animal position uncertainty.**
TOP: Top-down individual camera views, with dolphin bounding boxes in yellow (correlating boxes are numbered). The habitat-bisecting lines ($l_s$) for each camera frame are indicated in solid red. Distances from Bounding Box 2 (centered on the black and gray crosshair) to the closest frame boundary ($d_b$) and the boundary to the bisecting line ($d_l$) are indicated by the white measurement bars. MIDDLE: Meshed camera views including dolphin bounding boxes (yellow), with the location uncertainty distribution (A) overlaid for Box 2. BOTTOM: 2D location uncertainty distribution (A) with major (a-a, black) and minor (b-b, red) axes labeled and separately plotted.

for the other (far) camera as $w_f = 1 - w_n$. This ensured that if detection $u$ was on $l_s$, then $w_n = w_f = 0.5$, and as $u$ moved closer to $b_n$, we would have $w_n \rightarrow 0$ and $w_f \rightarrow 1$.

**False positive mitigation.** In specific circumstances, the shapes of the drains at the bottom of the habitat were warped by the light passing through rough surface water, and resulted in false dolphin detections. Separate (smaller) image classifiers for each camera were trained to identify these false positive drain detections, and were run on any detections that occurred in the regions of the video frames containing the drains. These detectors were strictly CNN image classifiers and were each trained on over 350 images and tested on over 150 images. For the drain detector, the input layer size had the format of $(l_d, l_d, 3)$, where $l_d$ is the mean side length of the detection bounding boxes being passed through the secondary classifiers. The feature detection layers had the same general structure as the Faster R-CNN classifier network, except in this case the convolution layers had, in order: 32, 48, 64, and 64 filters each. In the classification layers, the first fully connected layer had a length of 256.

## Temporal association of detections

Each experimental session involved the detection of multiple animals throughout their habitat. However, animal detections were done independently for each frame of the video. To extract kinematic information from the animals in the video, the detection associations needed to be preserved across frames. In this work, short continuous tracks (i.e. *tracklets*) were generated for a detected animal by identifying the most likely detection of that animal in the subsequent frame (Fig 3). To generate multiple individual tracklets in series of video frames, an iterative procedure of *prediction* and *association* was conducted under a Kalman filter framework with a constant velocity model.

The position of the $i$-th detected animal in one video frame at time $t$ is denoted as $\mathbf{u}^{(t,i)} = [u_x^{(t,i)}, u_y^{(t,i)}]$. Each detection, $\mathbf{u}^{(t,i)}$ was either associated with a currently existing tracklet or used to initialize a new tracklet. To determine which action was taken, for each tracklet, denoted as $\mathbf{T}^{(k)}$ for the $k$-th tracklet, this process first predicted the state of the tracked animal in the next frame ($\hat{\mathbf{T}}^{(k,t+1)}$) based on the current state information of the animal $\mathbf{T}^{(k,t)}$.

$$\mathbf{T}^{(k,t)} = [\mathbf{p}^{(k,t)}, \mathbf{v}^{(k,t)}] \tag{1}$$

$$= [p_x^{(k,t)}, p_y^{(k,t)}, v_x^{(k,t)}, v_y^{(k,t)}] \tag{2}$$

$$\hat{\mathbf{T}}^{(k,t+1)} = [\hat{\mathbf{p}}^{(k,t+1)}, \hat{\mathbf{v}}^{(k,t+1)}] \tag{3}$$

$$= [\hat{p}_x^{(k,t+1)}, \hat{p}_y^{(k,t+1)}, \hat{v}_x^{(k,t+1)}, \hat{v}_y^{(k,t+1)}] \tag{4}$$

where $\mathbf{p}^{(k,t)} = [p_x^{(k,t)}, p_y^{(k,t)}]$ denotes the filtered position of the animal tracked by the $k$-th tracklet at time $t$ and $\mathbf{v}^{(k,t)} = [v_x^{(k,t)}, v_y^{(k,t)}]$ is the corresponding velocity (Fig 3, popout-bottom). Under a constant velocity model, the predicted next frame position $\hat{\mathbf{p}}^{(k,t+1)} = [\hat{p}_x^{(k,t+1)}, \hat{p}_y^{(k,t+1)}]$ was obtained by integrating the current velocity over one frame period and summing this to the current frame position. The predicted velocity remained constant.

$$\hat{p}_x^{(k,t+1)} = p_x^{(k,t)} + v_x^{(k,t)} \Delta t \tag{5}$$

$$\hat{p}_y^{(k,t+1)} = p_y^{(k,t)} + v_y^{(k,t)} \Delta t \tag{6}$$

$$\hat{v}_x^{(k,t+1)} = v_x^{(k,t)} \tag{7}$$

$$\hat{v}_y^{(k,t+1)} = v_y^{(k,t)} \tag{8}$$

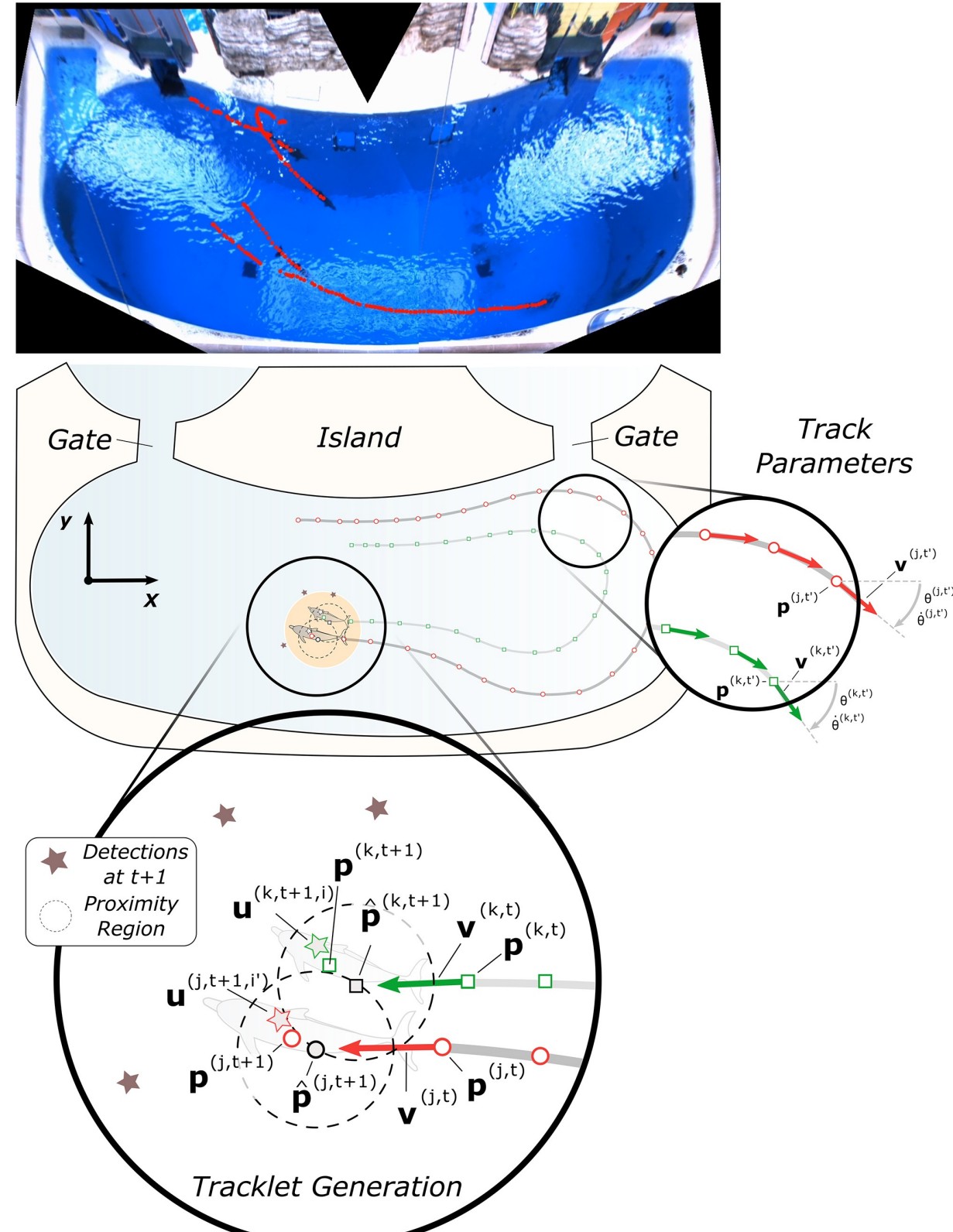

**Fig 3. Illustration of tracklet generation.** TOP: Tracklet segments (red) overlaid on a single video frame, generated by stitching the views from both cameras. Each tracklet in this frame was plotted from its inception to each corresponding dolphin's current position. While each dolphin can be

tracked, the lack of clarity when underwater impedes individual identification. CENTER: $x$-$y$ view of example tracklets (red and green on gray lines) of two dolphins (highlighted light orange), which are also shown in Fig 1, top. POPOUT-RIGHT: Vector illustrations of the two example tracks. Example notation for tracklet $j$ (red): position ($\mathbf{p}^{(j, t')}$), velocity ($\mathbf{v}^{(j, t')}$), yaw ($\theta^{(j, t')}$), and yaw rate ($\dot{\theta}^{(j,t')}$). POPOUT-BOTTOM Illustration of tracklet generation, with detections (stars) and tracklet proximity regions (dashed). Example notation for tracklet $j$ (red): position ($\mathbf{p}^{(j, t)}$), velocity ($\mathbf{v}^{(j, t)}$), Kalman-predicted future position ($\hat{\mathbf{p}}^{(j,t+1)}$), true future position ($\mathbf{p}^{(j, t+1)}$), and future animal detection ($\mathbf{u}^{(j, t+1, i')}$).

Using the predicted position, the $k$-th tracklet checked whether there existed a closest detection in the next frame that was within the *proximity region* of the predicted position, which is defined as a circle around the predicted position with radius 0.8 m (heuristically tuned). If true, that detection, denoted as $\mathbf{u}^{(k, t+1, i)}$ for the $i$-th detection in frame $t+1$ associated with the $k$-th tracklet, was used as the reference signal of the Kalman filter to update the state (position and speed) of tracklet $\mathbf{T}^{(k)}$. If false, the unassociated tracklet continued propagating forward, assuming the animal maintained a constant velocity. If a tracklet continued to be unassociated for 5 consecutive frames (empirically determined), it was considered inactive and was truncated at the last confirmed association. All information related to the $k$-th tracklet was saved after its deactivation:

$$\mathbf{T}^{(k)} = \left[ \mathbf{T}^{(k,t_{start})}, \cdots, \mathbf{T}^{(k,t-1)}, \mathbf{T}^{(k,t)}, \mathbf{T}^{(k,t+1)}, \cdots, \mathbf{T}^{(k,t_{end})} \right]^{T} \tag{9}$$

As illustrated in Fig 3, the tracklet formation operation linked each animal's individual detections ($\mathbf{u}$) over consecutive frames. This process returned the smoothed track positions ($\mathbf{p}$) of the animals, and by numerically differentiating the tracklets it was possible to extract the forward speed ($v$), yaw ($\theta$), and turning rate ($\dot{\theta}$), which could then be used to parameterize the positional states of the animals.

## Position uncertainty

There was a general position uncertainty for each animal detection due to noise in the Faster R-CNN detections. This was caused by a combination of limited camera resolution, as well as distortion of an animal's image from waves and ripples on the surface of the water. Additionally, since animal depth could not be measured, there were errors in the world-frame $x$-$y$ location estimates (caused by camera perspective and light refraction effects) that could not be corrected. This required a specialized $x$-$y$ position-dependent uncertainty distribution, based on prior knowledge of animal occupancy trends in the water column. Due to the high volume of data available to produce the underlying structure of the spatial distribution, the distribution kernels themselves could be directly generated rather than relying on estimation techniques.

In this work, the detection uncertainty was represented as a 2D probability density function (PDF), whose size and shape depended on the location of the detection with respect to the cameras (Fig 2, bottom, A). The short (minor) axis, $D_1$, was a Gaussian uncertainty distribution defined according to a heuristically estimated error in the camera detections ($\sim$0.2 m), and represented the general position uncertainty in the Faster R-CNN detections (Fig 2, bottom, b-b). The long (major) axis of the spatial distribution, $D_2$, represented the position uncertainty caused by the perspective and refraction effects (uncertainty from unknown depth). A 1D PDF was defined according to previously measured animal depth data (total of 9.8 hours during separate OTS time blocks), obtained via non-invasive tagging, which represented the general distribution of depths occupied by the animals. This was convolved with $D_1$ to produce the general shape of $D_2$ (Fig 2, bottom, a-a). The $x$-axis length scale for $D_2$ for a particular detection was obtained from the maximum position error in the detection's $x$-$y$ location. This was the magnitude of the $x$-$y$ position difference (original versus corrected $x$-$y$ position) if the

detection happened to be at maximum depth ($\sim$7 m). This magnitude varied dependent on the world-frame original location of the detection. Details on the depth-based location correction can be found in [25].

## Mapping animal kinematics to habitat

Heatmaps of dolphin position and speed were used to map animal positional state to the habitat. The dolphins were defined to be static or minimally mobile (drifting) when they were traveling at speeds below 0.5 ms$^{-1}$, and dynamic otherwise. To generate the positional heat maps, a blank 2D pixel map of the main habitat, $M$, was first created. Then, for each pixel representation $p$ of a detection $u$, the maximum possible magnitude of location error due to depth was determined, defined as $e_m$ (pixels, scale 1 pix = 5 cm), along with the orientation of the error propagation, $\psi_m$ (radians). The perimeter of the habitat served as a hard constraint on the location of the animals, thus $e_m$ was truncated if the location of the point with the maximum possible shift, $[p_x + e_m\cos(\psi_m), p_y + e_m\sin(\psi_m)]$, fell outside this boundary. The minor axis of the 2D spatial uncertainty distribution, $D_1$, was a 1D PDF in the form of a Gaussian kernel with $\sigma_{gauss}$ = 0.2$s$ (0.2 meters scaled to pixels by scaling factor $s$ = 20). Next, the depth PDF was interpolated to be $e_m$ pixels long, and was convolved with $D_1$ (to account for measurement uncertainty in the camera detections). This yielded the major axis 1D PDF, $D_2$. The 2D (unrotated) occupancy PDF, $E = D_1^\top D_2$, was then computed, where $D_1$, $D_2$ were horizontal vectors of the same length. The 2D rotated occupancy PDF, $F$, was calculated by rotating $E$ by an angle of $\psi_m$ through an interpolating array rotation. The MATLAB implementation of `imrotate` was used for this calculation. $F$ was then normalized to ensure the distribution summed to 1. Finally, $F$ was locally summed into $M$, centered at location $[x_u, y_u] = [p_x + 0.5e_m\cos(\psi_m), p_y + 0.5e_m\sin(\psi_m)]$, to inject the occupancy probability distribution for $u$ into map $M$. This process was then repeated for all detections. For the sake of visibility, all heatmaps were sub-sampled down to the scale of 1 pix = 1 meter.

A similar process was used to form the speed heatmaps. In a speed heatmap, the values of $F$ are additionally scaled by the scalar speed of the animal, $v$, that corresponds to detection $u$, and then locally summed into a separate map, $N$ (sum $F\cdot v$ into $N$ centered at $[x_u, y_u]$). Element-wise division of $N$ by $M$ was performed to generate $S$, a map of the average speed per location.

Lastly, the direction of motion of the animals throughout the monitored region was described using a quiver plot representation. To formulate the quiver plot, two separate heatmaps were generated, $Q_x$ and $Q_y$, one each for the $x$ and $y$ components of the animals' velocities. $Q_x$ was created using a similar method to the speed heatmap, but in this case $F$ was scaled by the $x$-component of the animal's velocity (sum $F\cdot v\cos(\theta)$ into $Q_x$ centered at $[x_u, y_u]$), where $\theta$ was the heading of the animal corresponding to detection $u$. Similarly for $Q_y$, $F$ was scaled by the $y$-component of the animal's velocity (sum $F\cdot v\sin(\theta)$ into $Q_y$ centered at $[x_u, y_u]$). The vector components $Q_x$ and $Q_y$ combined represented the general orientation of the animals at each point in the habitat.

## Probability distributions of metrics and entropy computation

For each time block of OTS and ITS, the PDFs of speed (ms$^{-1}$) and yaw (rad) were numerically determined. These were obtained by randomly extracting 10$^5$ data samples of both metrics from each time block of OTS and ITS, and producing PDFs for each metric and time block from these data subsets.

Additionally, the joint differential entropies of speed and yaw were computed for each time block of OTS and ITS. In this case, the joint entropy of animal speed and yaw represents the

coupled variation in these metrics for the animals. This indicates that speed-yaw joint entropy can be considered a proxy for measuring the diversity of their kinematic behavior. To compute the joint entropy $h$ for one time block, the randomly sampled speed (continuous random variable $\mathbf{S}$) and yaw (continuous random variable $\Psi$) data subsets ($S$ and $\Psi$, respectively) of that time block were used to generate a speed/yaw joint PDF: $f(s, \psi)$, where $s \in S$, $\psi \in \Psi$. $f$ was then used to compute $h$ with the standard method:

$$h(\mathbf{S}, \Psi) = -\int_{S,\Psi} f(s, \psi)\ln[f(s, \psi)]ds d\psi \tag{10}$$

## Kolmogorov-Smirnov statistics

To evaluate the statistical differences in animal dynamics between time blocks, the two-sample Kolmogorov-Smirnov (K-S) distances ($\Delta_{ks}$) and their significance levels ($\alpha$) were computed for each of the following metrics: speed (ms$^{-1}$), yaw (rad), yaw rate rads$^{-1}$), and the standard deviations of each [26]. These were done by comparing randomly-sampled subsets of each time block, with each subset consisting of $10^4$ data samples per metric. Only time blocks of similar type were compared (i.e. no ITS blocks were compared to OTS blocks, and vice-versa). K-S statistics were chosen to allow for nonparametric comparisons between probability distributions, as the metric distributions within each subtype (e.g. animal speed, yaw) did not all pertain to the same family of distributions (e.g. normal, exponential, etc.), rendering more traditional statistical comparisons ill-suited to this application. The computations were performed using the MATLAB statistics toolbox function `kstest2`.

## Results

### Detector and filter performance

During evaluation after training the networks, the Faster R-CNN detectors for the left and right cameras achieved Average Precision scores of 0.76 and 0.78, respectively. Additionally, during network training the CNN drain classifiers for the left and right cameras achieved respective accuracy scores of 92% and 94%. The performance of this pair of Faster R-CNN detectors with comparisons to ground truth was fully evaluated in [25], with the results reported in Table 3. To summarize the performance results: two additional monitoring sessions were video recorded and tracked both manually and using the automated CNN-based tracking system from this manuscript. During these sessions, two individual dolphins were tracked by a human observer and the results were compared to the detections produced by the automated system. Overall, for these two deployments the human tracker (representing the ground-truth) was able to detect the dolphins 88.1% of the time, while the CNN-based trackers

**Table 3. Performance comparison between manual and CNN animal detections for two sessions as part of a separate monitoring exercise, where individual dolphins were tracked as opposed to the entire group.** A1 and A2 refer to specific dolphins, with A1 being tracked over two recordings during Deployment 1, and A1 and A2 being tracked during the same recording during Deployment 2. "Detectability" is defined as the total time each individual dolphin was able to be detected by either the human or CNN trackers over each deployment period.

| Parameters | Deployment 1 | | Deployment 2 | | Overall |
|---|---|---|---|---|---|
| | **A1–1** | **A1–2** | **A1** | **A2** | |
| Duration [minute] | 22.6 | 30.9 | 48.9 | 49.2 | 151.6 |
| Detectability—Manual | 70.4% | 86.0% | 100% | 85.6% | 88.1% |
| Detectability—CNN | 44.6% | 54.7% | 50.2% | 59.2% | 53.2% |

were able to detect the dolphins 53.2% of the time. As a result, the automated system achieved an overall recall rate of 60.4% versus ground-truth.

Processing all 99.5 hours of recordings yielded $5.92 \times 10^6$ detections for the left camera and $6.35 \times 10^6$ detections for the right. The initial set of detections took $\sim 8.4$ days to compute when performed on the Titan V computer system. Of these, $3.83 \times 10^4$ (0.65%) detections from the left camera and $3.02 \times 10^4$ (0.48%) detections from the right camera were found to be drains misclassified as dolphins. After removing the misclassified detections, meshing the left and right detection sets yielded a total of $1.01 \times 10^7$ individual animal detections within the monitored habitat. The tracklet generation method used in this work associated animal track segments containing gaps of up to 4 time steps. As a result, the prediction component of its Kalman filter implementation was used to fill in short gaps in the tracking data. Generating tracklets from the meshed detections yielded a total of $1.24 \times 10^7$ estimated dolphin locations, from $3.44 \times 10^5$ total tracklets.

A note on detector limitations and the animal identification problem: while this system is robust in detecting a dolphin in-frame, it cannot track specific animals. The camera resolutions are not sufficient to resolve identifying features on the animals, and the environmental occlusions (glare regions, severe water surface disturbances) prevent continuous tracking (Fig 3, top). As a result, while each tracklet corresponds to a single dolphin at a time, the lack of identifiability prevents individual longer-duration tracking (>30 seconds) and therefore prevents individual metrics generation. For this reason, the results in this manuscript are presented for the dolphins as a group, rather than for each individual.

## Spatial distribution—Position

During OTS, the tracked animals were found to be in a dynamic swimming state $\sim 77\%$ of the time and a static state for $\sim 23\%$ of the time. The static OTS spatial distribution tended to be associated with particular features of their habitat: the gates that lead to the other areas of the habitat or at the underwater windows that offered views of the animal care specialist staff areas (Fig 4). When swimming dynamically during OTS, the dolphins tended to spend more time near the edges of their habitat, with the most time focused on the island side with the gates and the windows (Fig 5, left column). This was especially true during Block 5, with additional weight placed along the edge of the central island.

Throughout ITS, the dolphins were asked to engage in dynamic swimming tasks $\sim 62\%$ of the time, and were at station (in front of the ACSs) for the remaining $\sim 38\%$ of the time. During ITS, the dolphins had a heavy static presence in front of the central island, where the animals were stationed during formal training programs. The animals also spent less time around

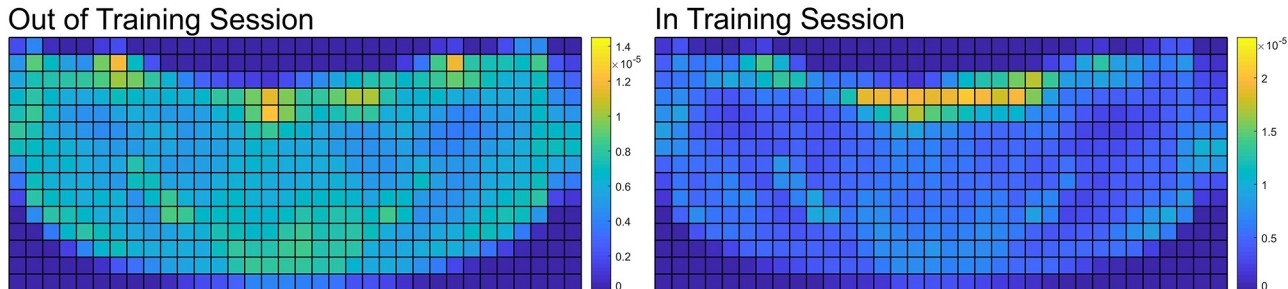

**Fig 4. Static position distributions for OTS and ITS.** A note on the format of the training sessions: Dolphins spent more time stationed at the main island during public presentations than non-public animal care sessions. During public presentations, ACSs spend a higher portion of the training session on the main island because it is within view of all of the public attending the presentation. Non-public animal care sessions are more fluid in their structure than public sessions. ACSs often use the entire perimeter of the habitat throughout the session.

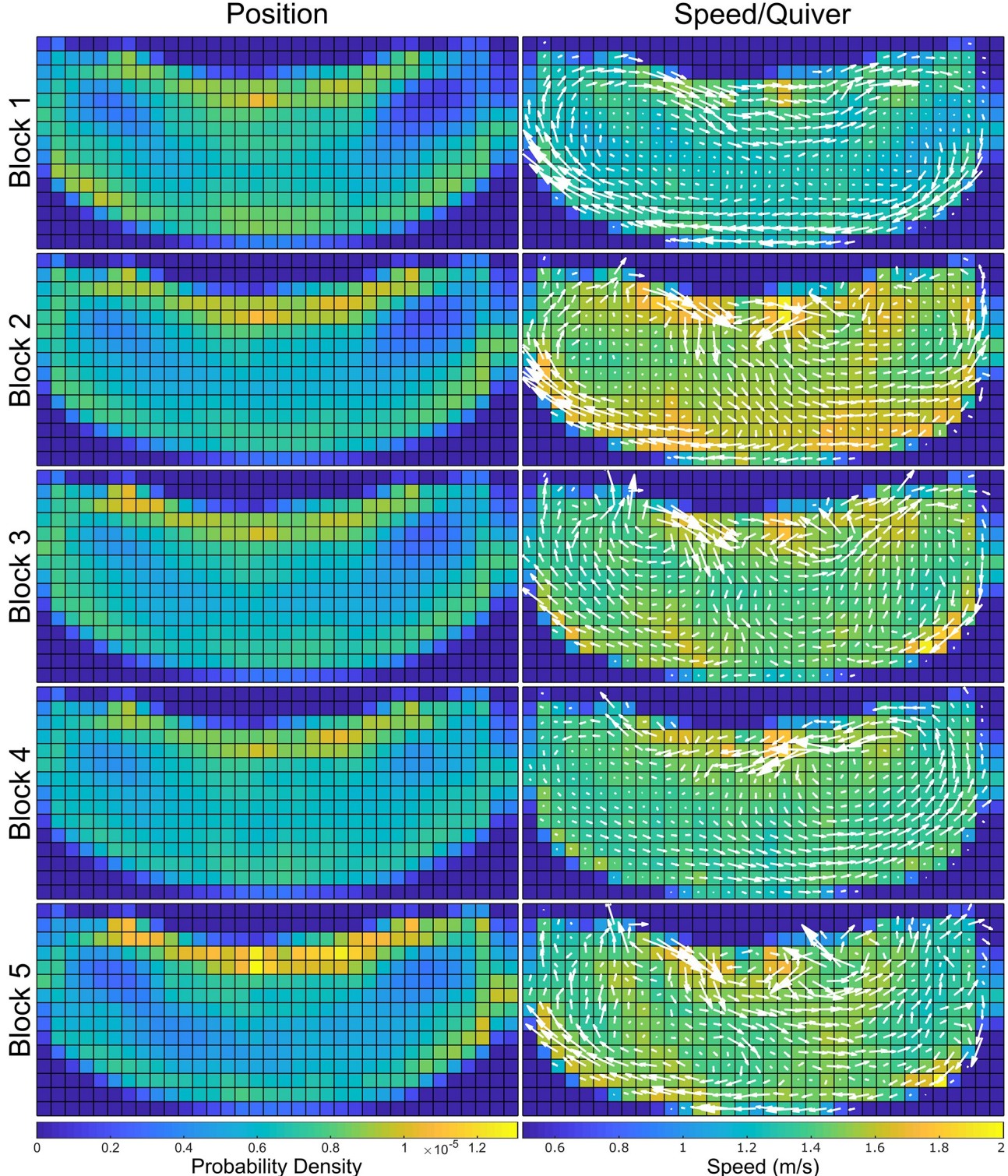

**Fig 5. Spatial distributions for dynamic OTS, with position distributions along the first column and speed distributions/quiver plots along the second column.** Prior to the first full training session of the day at 09:30, the dolphins were engaged in low intensity (resting) swimming clockwise around the perimeter of the habitat, with the highest average OTS speeds recorded after the 9:30 sessions. From there, speeds trail off for the subsequent two time periods. The 13:30–14:30 time block is characterized by slower swimming in a predominantly counterclockwise pattern. There is an increase in speed and varied heading pattern during the 15:00–16:00 time block.

the edges of the environment, in contrast with their locations during OTS (Fig 6, left column). During ITS, the ACSs presented discriminative stimuli or gestures corresponding to specific animal behavior, which defined the spatial distributions of the dolphins' movements during these time blocks. Additionally, there were spatial distribution similarities between training sessions of similar type, e.g. Blocks 1, 3 were animal care and husbandry sessions, and 2, 4 were public presentations. Note the structure of the spatial distributions across the top of their

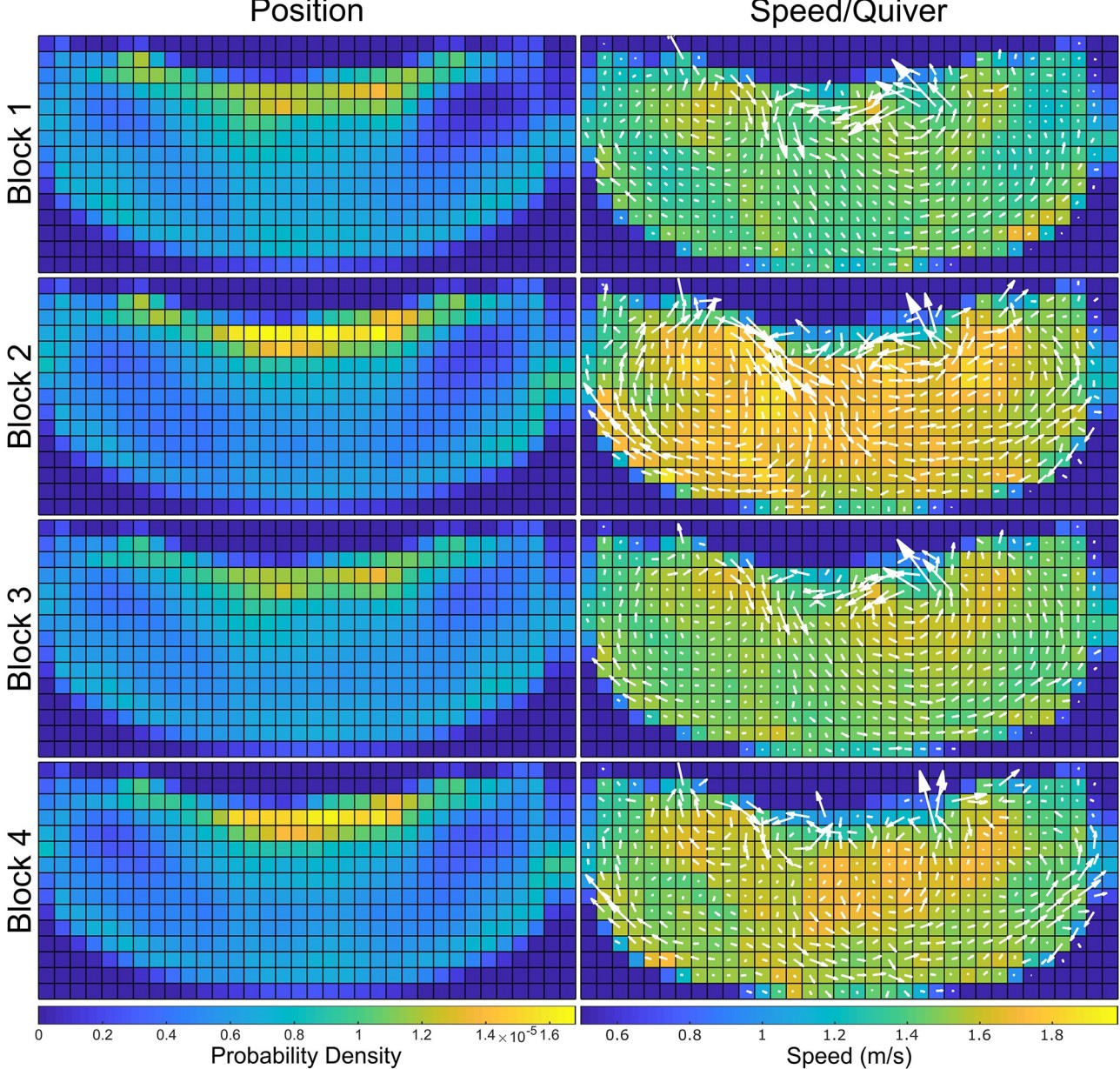

**Fig 6. Spatial distributions for dynamic ITS, with position distributions along the first column and speed distributions/quiver plots along the second column.** During the animal care sessions (Block 1: 09:30 to 10:00, Block 3: 13:00–13:30), the dolphins engaged in lower intensity swimming throughout the habitat than the presentation sessions (Block 2: 11:30–12:00, Block 4: 14:30–15:00). This difference is qualitatively explained through the discrepancy in ACS requests from the animals: high-intensity behaviors are prompted more often during presentations, while care sessions cover a wider variety of behaviors. Conversely, spatial coverage of the habitat does not have high variance within the ITS blocks, with an expectedly high concentration on the central island where the ACSs are located for all ITS blocks.

habitat, where during the care sessions (Blk. 1, 3) the dolphins' positions were focused on specific points in the area, while during the presentations (Blk. 2, 4) their positions were distributed across the edge of the central island. This captured the formation used during presentations with animals distributed more uniformly across the island. This also serves to qualitatively validate that the detectors are working as expected, given the dolphins are observed to be present in a region they are commonly instructed to occupy.

### Spatial distribution—Speed/quiver

In Block 1 of OTS, the dolphins had relatively low speeds (mean 1.30 ms$^{-1}$, Table 2) across their habitat, and based on the vector field of the quiver plot for the block, were engaged in large, smooth loops along the edges of the habitat (Fig 5, right column). This was contrasted with Block 2, which saw a higher general speed (mean 1.57 ms$^{-1}$) as well as diversified movement patterns, with the right half exhibiting counter-clockwise chirality while the left half maintained the clockwise motion pattern. Blocks 3–5 exhibited higher mean speeds than Block 1, and lower than Block 2 (Table 2), with the dolphins' movement patterns shifting changing between each OTS block (Fig 5). In contrast, there was no such pattern in the dolphins' mean yaw rates (Table 2).

During ITS, the care blocks' (Blk. 1, 3) speed distributions and vector fields qualitatively demonstrated similar structures, while those of the presentations (Blk. 2, 4) were more mixed, with more similarities along the left and right far sides, but fewer in the center (Fig 6, right column). The mean speeds and mean yaw rates did not share particular similarities between blocks of similar type (Table 2). In general, speeds across the entire habitat are higher during public presentations than non-public animal care sessions because high-energy behaviors (e.g., speed swims, porpoising, breaches) are typically requested from the dolphins several times throughout the presentation. Though non-public presentations include high-energy behaviors, non-public animal care sessions also focus on training new behaviors and engaging in husbandry behaviors. Public presentations provide the opportunity for exercise through a variety of higher energy behaviors, and non-public sessions afford the ability to engage in comprehensive animal care and time to work on new behaviors.

### Joint entropy results for kinematic diversity analysis

The joint differential entropies of speed and yaw per time block are displayed in Fig 7, bottom, with values reported in Table 4. The time blocks in this figure are presented in chronological order, and we observed the lowest kinematic diversity in the mornings (the first blocks of each OTS and ITS) as the animal care specialists were arriving at work and setting up for the day. The highest kinematic diversity when not interacting with the ACSs then occurred immediately after the first ITS time block. In general, the first time blocks of both OTS and ITS showed the lowest kinematic diversity of their type, the second of each showed the highest, and the following blocks stabilized between the two extremes. The speed/quiver plots (Figs 5 and 6, right) provide a qualitative understanding of the entropy results. For example, in Block 1 of OTS (Fig 5, top-right) the dolphins engaged in slow swimming throughout their habitat in smooth consistent cycles along the environment edge, yielding the lowest joint entropy. Joint entropy then increased during both the morning ITS and OTS blocks and remained elevated for the rest of the day, representing higher animal engagement through the middle of their waking hours.

### Statistical comparison of metrics for behavior differentiation

The K-S statistics were used to confirm the similarities and differences between time blocks within both OTS and ITS. To aid in visualizing this, Fig 7, top, displays the overlaid PDFs of

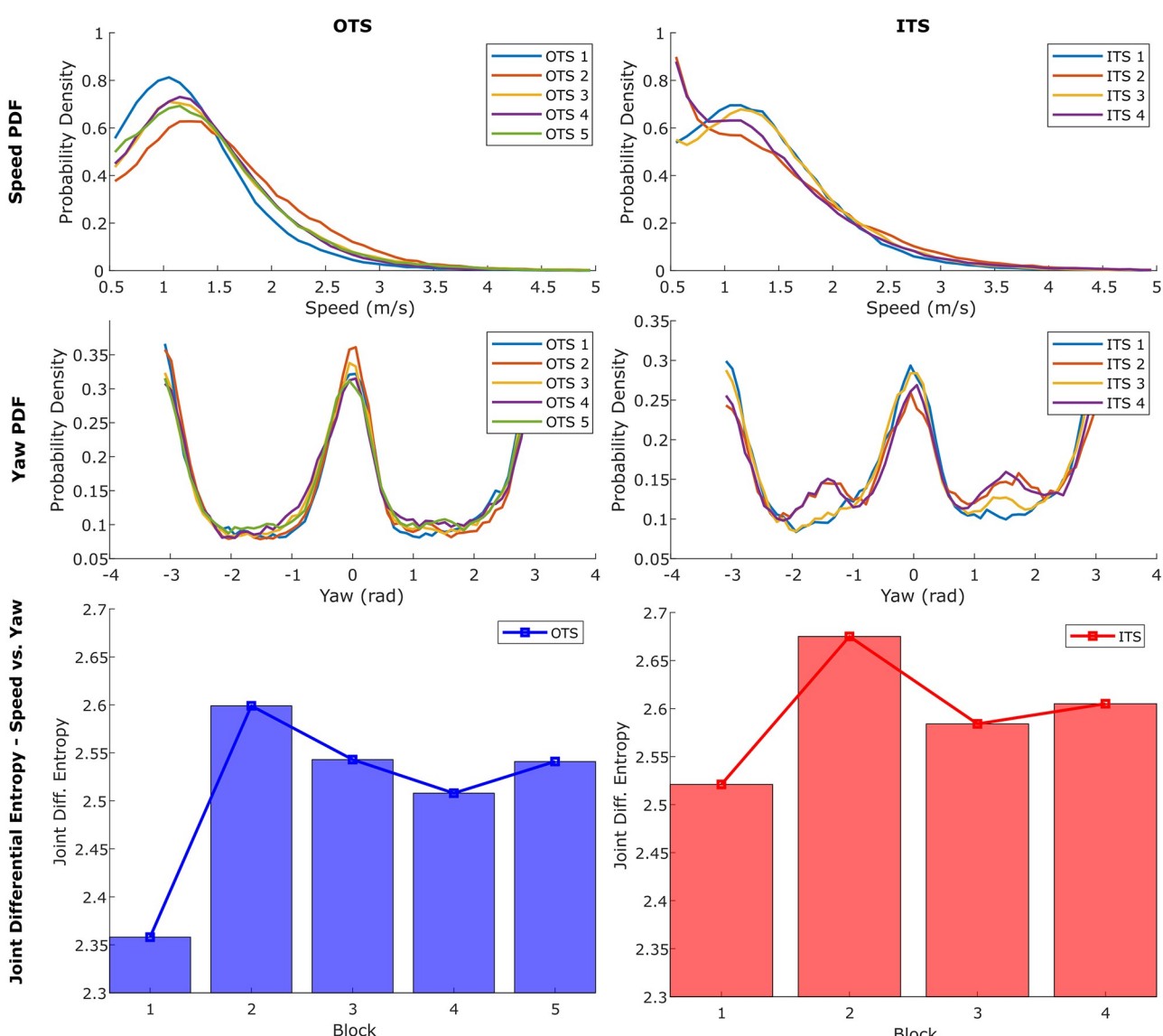

**Fig 7. Speed and yaw probability distributions and joint differential entropies, respective to time block.** TOP: Probability density functions of animal speed (m s$^{-1}$) for OTS (left) and ITS (right). MIDDLE: Probability density functions of yaw (rad) for OTS (left) and ITS (right). BOTTOM: Joint differential entropy of speed and yaw for each block of OTS (left) and ITS (right), with limited-range y-axes to more clearly show value differences.

the speed and yaw metrics during OTS, and Fig 7, middle, displays the PDFs during ITS. A complete table with K-S distances and $\alpha$ values for all six metrics is present in S1 Table in the supporting information, with all values rounded to 3 digits of precision. For OTS, we saw from the K-S results that Blocks 1 and 2 varied the most with respect to the others in terms of speed, which was observed in Fig 7, top, while the yaw values were not generally significantly

**Table 4. Speed and yaw joint differential entropy.**

| Block | OTS | | | | | ITS | | | |
|---|---|---|---|---|---|---|---|---|---|
| | 1 | 2 | 3 | 4 | 5 | 1 | 2 | 3 | 4 |
| **Entropy** | 2.358 | 2.599 | 2.543 | 2.508 | 2.541 | 2.521 | 2.675 | 2.584 | 2.605 |

different, again observed in Fig 7 (given the high number of samples used to generate the K-S statistics, we were able to compare the significance levels to a stronger threshold of $\alpha_{crit} = 0.001$). Across the board, Block 2 generally differed significantly from the rest of the OTS blocks for the most metrics, with Block 1 following close behind. In contrast, Blocks 3–5 differed the least significantly from each other, indicating similarities in the dolphins' dynamics patterns for Blocks 3–5.

For ITS, we note that the significant differences in metrics generally followed the structure type of each ITS block: comparisons between Blocks 1 vs. 3, and 2 vs. 4, were found to be significantly different the least often. As the ACSs requested similar behaviors during ITS blocks of the same type, we expected similarities in the dynamics metrics for Blocks 1 vs. 3 (animal care sessions) and Blocks 2 vs. 4 (presentations), and differences between the metrics for blocks of different types. Of particular note are the yaw std. dev. and yaw rate std. dev. metrics, with entire order of magnitude differences in K-S distances when comparing similar vs. different types of ITS blocks. Overall, the pattern displayed by the ITS K-S statistics in S1 Table correlated with this expectation.

## Discussion

### Automatic dolphin detection

This research presents a framework that enables the persistent monitoring of dolphins under professional care through external sensing, performed on a scale that would be prohibitive for traditional manual tracking. Both the Faster R-CNN dolphin detection and CNN drain detection methods displayed reliable performance in testing, and enabled large-scale data processing at rates not achievable by humans. Given that the total duration of video processed was $\sim$199 hours (2 cameras $\times$ 99.5 hours each), an inference time of $\sim$202 hours (1.013$\times$) represents at minimum an order-of-magnitude increase in processing speed when compared to human data annotation. This estimate was obtained from the authors' prior experience in manual animal tracking, which could take over 10 hours of human effort per hour of video (frame rate of 10 Hz) annotated for a *single* animal. In this research, the detections generated by the monitoring framework were used to estimate two-dimensional animal position and kinematics (speed, heading) to characterize animal behavior and spatial use within their environment. As such, this detection framework presents new opportunities for long-term monitoring of animal kinematics, and enables the automated processing of the longer duration and more frequent recording sessions that will provide a more complete picture of animal behavior in these environments.

### Animal kinematics and habitat use

**Kinematic diversity.**  Joint dynamic entropy was used to quantify differences in animal kinematic diversity throughout the day to explore how temporal changes in the dolphins' habitat would result in modified kinematic diversity levels (Fig 7, bottom). The use of entropy as a proxy for kinematic diversity has been applied in the past to characterize prey motion unpredictability for predator evasion, however in this work it serves to provide a measure of animal engagement [27]. The kinematic diversity results presented here are consistent with previous research on animal activity and sleep patterns, which reports a diurnal activity cycle for animals under professional care [12]. However, it is interesting to note that changes in animal kinematic diversity throughout the day during OTS are not gradual: the OTS time block displaying the minimum value is immediately followed by the block displaying the maximum, and are only separated by the first training session (30 minute duration). This sudden shift may not be fully explained by only the dolphins' diurnal activity cycle, and may be related to

the fact that their first daily interactions with the ACSs occur between these two OTS time blocks. A finer time-scale analysis of their kinematic diversity trends is necessary to determine which is the cause for this change in animal engagement.

**Habitat use.**   The kinematic data also enabled an investigation into how features in the habitat influenced animal behavior and spatial use, particularly during OTS. The animals tended to have a general focus on the area between the gates along the edge of the central island (Fig 5, left). Additionally, throughout the OTS position plots (including static, Fig 4, left) four animal-preferred locations were observed. The two hot spots to the left and right of the central island are gates (Fig 1, bottom), where the dolphins could communicate with conspecifics when closed or pass through to other areas of their habitat when open. Conversely, the two hot spots nearer the middle of the island edge corresponded to underwater windows that led to an ACS work area (two central windows in Fig 1, bottom). Through these windows the dolphins may observe the ACSs, view conspecifics in one of the back habitats (through an additional window, not shown in Fig 1), or observe enrichment occasionally placed on the other side of the glass (mirrors, videos, etc.). Regions of the habitat in proximity to these two windows experienced some of the highest occupancy in all OTS position plots, both static and dynamic. This indicates that particular attractors for the dolphins' attention were observable through those windows, whether they were the ACSs, conspecifics, or enrichment.

These attractors were also correlated with the dolphins' kinematics and activity levels. Of all the regions in the environment, only the positions in front of the central windows consistently recorded peak or near-peak location-specific animal swimming speeds for all OTS time blocks (Fig 5, right). When combined with the results from the spatial distributions (Fig 5, left), this implies that these dolphins not only focused their attention on these regions, their presence correlated to higher activity levels in the dolphins when swimming in their vicinity.

**Behavior classification from dynamics metrics.**   During ITS blocks, ACSs asked for specific behaviors from the dolphins and these behaviors were often repeated. Elements of public educational presentations (ITS 2/4) were varied to include a mixture of both high and low energy segments, and this blend resulted in similar dynamic patterns for the public sessions. In contrast, the non-public animal husbandry and training sessions (ITS 1/3) were less dynamic overall, and yielded similar dynamics patterns within these types of sessions. Qualitative similarities in the pairs of animal training sessions were observable in both the position and speed/quiver plots in Fig 6, and the probability density functions presented in Fig 7. Along with the statistical observations in S1 Table, without prior knowledge of the block types it would be possible to use this pattern to identify that Blocks 1 and 3 were likely the same type, as were 2 and 4. This demonstrates that the presented method of obtaining and analyzing the dolphins' dynamics metrics was has the potential to differentiate between general behavior types.

This was useful for analyzing the OTS results, as the position and speed/quiver plots in Fig 5 only showed patterns in the animals' location preferences within their habitat. In contrast, an analysis of the K-S results allowed for the identification of the statistical differences in animal dynamics between OTS time blocks. Block 2 separated itself significantly from all other time blocks in nearly every metric, while Block 1 was in a similar position (though not as pronounced). Blocks 3–5 showed few significant differences for metrics comparisons between each other. This indicated that the dolphins had more distinct dynamics for Blocks 1 and 2, and maintained similar dynamics patterns throughout Blocks 3–5. When combined with the joint differential entropy values, these results indicated there may be three general OTS behavior types for the dolphins in this dataset (in terms of kinematic diversity [KD]): "Low KD" at the beginning of the day (Block 1), "High KD" immediately after the first training session (Block 2), and "Medium KD" for the remainder of the day (Blocks 3–5).

## Limitations and future work

Using a limited number of cameras meant full stereo coverage of the habitat was not possible, preventing a direct estimate of animal depth. Additionally, camera placements resulted in region-specific glare on the surface of the water that impeded the Faster R-CNN detector. To address these problems, cameras could be added to locations that allow for fully overlapping coverage, at angles that avoid glare in the same regions. Further, installing cameras capable of low-light recording could enable night monitoring sessions. An inherent problem with camera-based tracking is the fact that similarities between dolphin profiles make it challenging to identify individuals. This problem has been addressed in [25], where kinematic data from dolphin-mounted biologging tags were used to filter camera-based animal location data. This filtering process made it more feasible to identify which location data points corresponded to specific tagged individuals, coupling the kinematic and location data streams for these animals. Fusing the coupled tag and camera data through methods similar to [25] or [28] would then provide high-accuracy localization information to contextualize the detailed kinematics data produced by the tags.

Beyond technical improvements, the next step in this research is to use long-term animal monitoring to inform management policy and aid welfare tracking. By working closely with the ACSs at the Brookfield Zoo, we intend to use the techniques presented in this manuscript to observe animal interactions with conspecifics and enrichment over time, track activity levels, and measure the effects of changing environmental conditions (e.g. effects of varying crowd size, spectator presence). In particular, given the emphasis the dolphins placed on particular regions of the environment, it will be important to evaluate the effects of attractors in these areas by varying enrichment type, physical placement, duration/time of exposure, and by recording ACS presence and interactions within these areas. In this way, we aim to guarantee a high level of animal engagement and work to identify potential stressors that may aid the Zoo in caring for their dolphins.

Further, the use of unmanned drones and gliders has the potential to extend this research for implementation in a wild setting. CNN tracking is already at work in whale [29] and shark [30] tracking in the wild, and the inclusion of these vehicles will open up new opportunities in making this research physically portable. The methods in this manuscript, particularly the tracklet generation, can be useful for not only identifying and localizing the animals, but also in providing basic kinematic information on entire groups, which is not generally feasible with tagging operations due to the limited number of tags available for deployment.

## Conclusions

Through this research we have demonstrated a monitoring framework that significantly enhances the efficiency of both data collection and analysis of dolphin movement and behavior in a zoo setting. This work demonstrated the feasibility of a camera-based computer-automated marine animal tracking system, and explored its capabilities by analyzing the behavior and habitat use of a group of dolphins over a large time scale. From the results, we were able to quantify day-scale temporal trends in the dolphins' spatial distributions, dynamics patterns, and kinematic diversity modes. These in turn revealed that habitat features associated with particular attractors served as focal points for this group of dolphins: these features were correlated with higher animal physical proximity, kinematic diversity (specifically ACS presence), and activity levels.

## Supporting information

**S1 Table. Kolmogorov-Smirnov session comparison.**
(PDF)

## Acknowledgments

The authors would like to thank the Brookfield Zoo for its aid in facilitating this research. Rita Stacey and the Seven Seas Animal Care Specialists were instrumental in helping to acquire such a large volume of data, and the help of the Zoo's administration made this research a possibility. Finally, the authors would like to thank Sarah Breen Bartecki and William Zeigler of the Chicago Zoological Society for their continued support.

## Author Contributions

**Conceptualization:** Joaquin Gabaldon, Lisa Lauderdale, K. Alex Shorter.

**Data curation:** Joaquin Gabaldon, Ding Zhang, Lisa Lauderdale.

**Formal analysis:** Joaquin Gabaldon, Ding Zhang, Lisa Lauderdale, K. Alex Shorter.

**Funding acquisition:** Lance Miller, K. Alex Shorter.

**Investigation:** Joaquin Gabaldon, Ding Zhang, Lisa Lauderdale.

**Methodology:** Joaquin Gabaldon, Ding Zhang, Matthew Johnson-Roberson, Kira Barton, K. Alex Shorter.

**Project administration:** Lance Miller, K. Alex Shorter.

**Resources:** Joaquin Gabaldon, Ding Zhang, Lisa Lauderdale, K. Alex Shorter.

**Software:** Joaquin Gabaldon, Ding Zhang.

**Supervision:** Lisa Lauderdale, Lance Miller, Kira Barton.

**Validation:** Joaquin Gabaldon, K. Alex Shorter.

**Visualization:** Joaquin Gabaldon, Ding Zhang, Kira Barton, K. Alex Shorter.

**Writing – original draft:** Joaquin Gabaldon, Ding Zhang, K. Alex Shorter.

**Writing – review & editing:** Lisa Lauderdale, Lance Miller, Kira Barton.

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
