## [Decision Letter · Decision Letter 0]

28 Oct 2021

PONE-D-21-19199Vision-based monitoring and measurement of bottlenose dolphins' daily habitat use and kinematicsPLOS ONE

Dear Dr. Gabaldon,

Thank you for submitting your manuscript to PLOS ONE. After careful consideration, we feel that it has merit but does not fully meet PLOS ONE’s publication criteria as it currently stands. Therefore, we invite you to submit a revised version of the manuscript that addresses the points raised during the review process. First, I must apologize to the authors for the delay on my decision. It was very difficult securing reviewers for this manuscript. This manuscript has now been reviewed by three experts in the field, and all three generally agree that the manuscript is technically quite sound. The reviews did provide a large number of comments, but these were mostly aimed at making the text more clear. Please thoroughly address all of these comments when submitting your revised manuscript.

We look forward to receiving your revised manuscript.

Kind regards,

William David Halliday, Ph.D.

Academic Editor

PLOS ONE

Journal Requirements:

Reviewers' comments:

Reviewer's Responses to Questions

**Comments to the Author**

1. Is the manuscript technically sound, and do the data support the conclusions?

Reviewer #1: Yes

Reviewer #2: Yes

Reviewer #3: Yes

2. Has the statistical analysis been performed appropriately and rigorously? 

Reviewer #1: I Don't Know

Reviewer #2: Yes

Reviewer #3: Yes

3. Have the authors made all data underlying the findings in their manuscript fully available?

Reviewer #1: Yes

Reviewer #2: No

Reviewer #3: Yes

4. Is the manuscript presented in an intelligible fashion and written in standard English?

Reviewer #1: Yes

Reviewer #2: Yes

Reviewer #3: Yes

5. Review Comments to the Author

Reviewer #1: This is a well-written manuscript that presents primary scientific research in an intelligible fashion and has been written in standard English. This research uses a dual camera system to record the study area and uses deep learning computer techniques (Convolutional Neuron Networks) to detect dolphin presence and movements within the frame of view (FOV). The authors used this information to describe habitat use and calculate speed throughout the environment during different time periods of the day (blocks) and whether the dolphins were in training session (ITS) or out of training session (OTS). Habitat use was described with heatmaps and by associating environmental features such as enrichment, windows and trainer, though no statistical analyses was presented to support these assertions. Speed was velocity (fluking speed m.s-1) and was used to describe how the dolphins moved throughout the environment by being condensed into static or dynamic movement. Kolmogorov-Smirnov tests were used to independently test the differences between blocks during OTS or ITS. Independently analysing the results of ITS and OTS is valuable as during the ITS dolphins are being asked to perform specific behaviour and therefore, differences between their OTS time would be expected. More clarification around the statistics used in this manuscript are warranted as the authors have not justified their use of the Kolmogorov-Smirnoff test over other available statistics or provided adequate details on the development and use of their heatmap method. This manuscript appears to present original research and the authors have made their data accessible.

The authors demonstrate that the use of deep learning computer techniques is achievable for video monitoring animals within a managed facility, and provide a good discussion of how the speed of analysis is greatly improved in comparison to manual video analyses. These conclusions are supported by the results supplied in this manuscript. Habitat use information was well-presented and reasoned, however, caution should be used when interpreting the conclusions as there was no formal test showing differences in use and environmental features. Authors state that day-scale temporal trend were able to be detected, this seems like a fair statement due to the timing of when video data was taken.

Page 1 Line 8: Authors state that “there is a strong emphasis on behavioral monitoring to inform welfare practices” but do not mention this again. Is there a suggestion that the continual video monitoring and use of CNN methods would be applicable to monitor behaviours linked to the welfare of these animals in the future? If so, the authors should provide greater detail into how these methods could benefit behavioural analyses

Page 3 Line 75: Authors provide an average age of the seven bottlenose dolphins in this study (17 +/- 12). Due to the large range in ages I do not believe this metric is a good descriptor of dolphin ages and authors should simply provide the ages.

Page 8 Line 273 onwards: In this section the authors provided details on their heatmap production method. Supplying the software that was used in generating these heatmaps would be beneficial to readers and for reproducibility. Additionally, have these methods been described previously in other literature, or is this a method that was developed by the authors? Either way, authors should state how and why they used these methods when other commonly used methods exist (for example kernel density estimation, which can have a barrier function for use with hard limits to movement, such as the walls of a pool).

Page 10 Line 325 onwards: Justification of the use of Kolmogorov-Smirnoff non parametric tests are warranted in this section. A statement of the assumptions and how the collected data has met these assumptions is important for assessing the appropriate use

Page 14 Line 486 onwards: The linking of dolphin distribution to environmental features, such as gates and windows has sound logic, however, no formal quantitative tests were performed. The links made between these features and kinematics supports these assumptions but it is difficult to tell if this is a correlation or driver of habitat use. The authors could consider analysing the differences in habitat use using species distribution models.

General comments:

1. The authors focus their conclusions around the enhanced ability these methods provide to efficiently analyse video data. These conclusions are in line with regards to the results, however, it would be of interest to hear how the results and the methods presented in this manuscript may be applicable to different systems. With the increased use of drones, video data is more frequently being collected for wild cetaceans, can CNN computational methods aid in analysing this data and what are the benefits or potential uses of using the methods presented in this manuscript.

2. Through describing a method that enables a more efficient analysis of video monitoring data, the authors have also created a greater understanding of the distribution (and potential environmental reasoning) for bottlenose dolphins present in this captive environment.

a. Are the difference observed in kinematics a natural occurring behaviour, a function of the environment, or

due to different stimuli occurring at different times of day?

b. If the dolphins are spending the majority of their OTS at areas where more enrichment is occurring

(windows, gates etc.) are there suggestions the authors could make about how these animals are

managed?

Expanding on the habitat use and kinematic results is relative because the title and points made throughout the manuscript suggest that habitat use and kinematics is the main focus and result. If this point is true than more emphasis should be place on discussing the implications of these findings for these dolphins. However, upon reading the manuscript, the impression is that the authors are describing new methods for analysing video data, and that habitat use and kinematics are potential uses for this tool. If this latter point is true, then the authors should consider changing the title to suggest a methods manuscript e.g. “Application of CNN in analysing video-based monitoring data for daily habitat use and kinematics of captive bottlenose dolphins”.

3. A point of interest, the kinematics and habitat use of bottlenose dolphins was provided as a group summary, rather than for individuals. Do all dolphins consistently group together and follow the same path while in the enclosure, or, is there individual variation present. Can CNN analysis techniques detect and follow individuals through time? If the authors could provide comment on this I believe it would, at least, be a valuable discussion point as a future direction for this work.

4. The authors have provided an ethics statement that lists the names of two organisations that approved this research. Do they have an ethics/permit number that could be referred to, that they could supply?

Reviewer #2: This study describes the implementation of a camera based auto-tracking approach to monitor dolphin locomotion in a managed area. The approach described is sound and the results suggest that it is effective for monitoring animal behavior, with the possibility to expand or enhance its performance through additional cameras or sensors. This work is consistent with other auto-tracking programs that have been described recently (reviewed in Panadeiro et al, 2021; https://doi.org/10.1038/s41684-021-00811-1), but is focused on the specific application of dolphins in an managed enclosure. Overall, I don't have any major concerns regarding this study. It is clear and well written, and will likely be of interest to scientists working in this area. One minor comment: On line 101, formal training sessions are defined as ITS, but under Table 2, it states that OTS blocks 2 & 4 are formal presentations. What is the difference between formal training and formal presentations? Perhaps consider using more distinct language to describe these?

Reviewer #3: This article struggles between two possible narratives:

(1) a new ML-based method to track dolphins in captivity, with data to show the utility of the methods, and

(2) a dolphin distribution and behavior study in captivity that uses a ML-based method to compare behaviors while in two different behavioral conditions.

I believe that the goal of the authors is the first narrative and as such, my comments below are reflective of this.

Abstract

- Lacks a "so what" or big picture. Why should someone want to read this article? What is the new innovation that helps to advance monitoring of dolphins in captivity?

Introduction

- This introduction does not prepare the reader for what is to come in the article and lacks reference to the body of work done with dolphins in captivity and in machine learning detection.

- The statement "Here we present an automated..." in paragraph 1 comes too soon. You haven't even described any of the previous research in the field. This should be in your last paragraph.

- The paragraph about tags in the wild "Biomechanics and behavioral studies..." does not add anything to the article and is not relevant given that the entire study is done in captivity and there is no comparison or attempt to implement this in the wild.

- I did not feel like the last paragraph that describes this study is accurate. It did not prepare me for what was to come. Please make sure they match in content and breadth as well as order ideas are presented in methods.

- Suggested structure: paragraph 1 = what we've previous learned from behavior studies of captive animals, paragraph 2 = methods used to study captive animals and limitations, paragraph 3 = overview of machine learning detection algorithms showing other applications of RPN and Fast R-CNN (since you are not inventing them in this study) and remaining gaps, paragraph 4 = here's what we are going to do and show in this article

Methods

- Overall, methods are not organized in a manner that the reader feels like they build on each other. We switch from behavior related things to ML related things. Too many details on the behavior stuff given that its not the focus.

- First paragraph in M&M feels unnecessary and almost more appropriate to do in intro.

- None of the details about the tank feel necessary nor relevant as they are really never revisited. Maybe add these measurements to the top panel of figure 1 and then there is no need to repeat them in text.

- Figure 1 and 2 are way too busy and need to be broken into different panels. I would combine top panel of fig1 and fig 2 as 1 figure. Tracking algorithm its own figure and prob dist their own as well. Label panels (a), (b), (c) as top, bottom etc. didn't really help because there were so many panels in each figure.

- Table 2. Add your speed, yaw metrics in this table with the time intervals.

- What are the metrics of detector performance?

- False positives should be their own section

- Tracklet is first used in title so doesn't lend well to preparing reader. Use this term in intro when describing what is going to be done.

- Should reference tracklet figure earlier on in this section.

- What is proximity region? How was it defined?

- Heatmap representation is really confusing given the aims of the study to track. Why isn't an individual track shown? This would be more powerful representation that the left column panels in figure 4 and 5.

- Suggested structure: section 1 = camera setup in tank area + how much data was collected over what periods of time, section 2 = manual labelling and analysis, section 3 = neural net description + training + metrics of detect performance, section 4 = detection processing to combine frames, 5 = position uncertainty, 6 = tracklets + heatmaps to visualize, 7 = drain detector, 8 = about different behavior states and dolphins training sessions

Results

- Given that this article seems to focus a lot on the detector and tracklet, there is only a single paragraph describing the performance of these algorithms and most of it is about the detector. How was the tracklet algorithm performance? What there any human ground-truthing of animal trajectory to confirm the tracklet trajectories?

- Make a table to summarize performance of algorithms.

- The comparison between OTS and ITS feels odd given that it is framed as comparing behavioral states and locations of the animal. In ITS, it is really obvious that the dolphins are going to be where the algorithm finds them in front of the stage area because that is where the trainer tells them to be. So it really isn't a "distribution" or their space use in the tank. I would frame this more as a confirmation of your detector working because the detector found the dolphins where they are supposed to be and maybe should be included in detector performance

- Figure 4 and 5. The differences in the space use don't really stand out with these plots. It may be best to plot a single track of an animal in OTS and one in ITS to show the space use. There's also a ton of text in the figure captions that isn't mentioned in the main text.

- Table 3 should be in the supplementary.

- I don't think yaw is ever actually defined or described.

- Statistical comparison and entropy should not be their own sections but rather woven into the Speed + Yaw description of the animals.

- Suggested structure: section 1 = detector performance, 2 = tracklet performance, 3 = animal space use during OTS and ITS, 4 = speed + yaw of animals

Discussion

- Most of the paragraphs in the discussion read as results paragraphs and should be moved to the results. These paragraphs explain the results so much more clearly than the results section. I really didn't comprehend the results of the article until I read the discussion section.

- section kinematic diversity = 1st paragraph reads like a results paragraph

- section behavioral classification = where these results even mentioned in the results section? it feels very unclear

- Order the kinematic diversity, habitat use and behavioral classification in the same order as the results. (i.e., habitat use, behavioral classification and kinematic diversity??)

Overall Comments:

- Distribution is not the same as behavior and it feels like these are used interchangeably throughout. When you are talking about the position of the animal, you are referring to their distribution within the tank. As such, the word habitat use should also not be used as a tank is not a habitat. Space use in tank would be more appropriate. Behavior refers to the speed, yaw and dynamic swimming state.

- Statistics confirm that there are real, statistical differences and similarities in data. They aren't used to "quantify" them or "give clearer view". Be careful of how you use the KS statistics in the discussion. Really, they just tell you that the patterns, differences and similarities that you see are real and not due to lack of sample size.

- Using the word "managed" is confusing as there are many wild populations that are managed. Use the word "captivity". Make it clear to the reader that this sort of study is only possible in captivity. This setup would not work in the wild.

6. PLOS authors have the option to publish the peer review history of their article (what does this mean?). If published, this will include your full peer review and any attached files.

Reviewer #1: No

Reviewer #2: No

Reviewer #3: No

---

## [Author Response · Author response to Decision Letter 0]

10 Jan 2022

Dear Dr. William Halliday,

Thank you for the thorough and thoughtful review of our manuscript by the associated editor and the reviewers. We have examined the reviewers’ comments carefully and made appropriate changes to the text as outlined below.

Reviewer #1

General Comments:

This is a well-written manuscript that presents primary scientific research in an intelligible fashion and has been written in standard English. This research uses a dual camera system to record the study area and uses deep learning computer techniques (Convolutional Neuron Networks) to detect dolphin presence and movements within the frame of view (FOV). The authors used this information to describe habitat use and calculate speed throughout the environment during different time periods of the day (blocks) and whether the dolphins were in training session (ITS) or out of training session (OTS). Habitat use was described with heatmaps and by associating environmental features such as enrichment, windows and trainer, though no statistical analyses was presented to support these assertions. Speed was velocity (fluking speed m.s-1) and was used to describe how the dolphins moved throughout the environment by being condensed into static or dynamic movement. Kolmogorov-Smirnov tests were used to independently test the differences between blocks during OTS or ITS. Independently analysing the results of ITS and OTS is valuable as during the ITS dolphins are being asked to perform specific behaviour and therefore, differences between their OTS time would be expected. More clarification around the statistics used in this manuscript are warranted as the authors have not justified their use of the Kolmogorov-Smirnoff test over other available statistics or provided adequate details on the development and use of their heatmap method. This manuscript appears to present original research and the authors have made their data accessible.

The authors demonstrate that the use of deep learning computer techniques is achievable for video monitoring animals within a managed facility, and provide a good discussion of how the speed of analysis is greatly improved in comparison to manual video analyses. These conclusions are supported by the results supplied in this manuscript. Habitat use information was well-presented and reasoned, however, caution should be used when interpreting the conclusions as there was no formal test showing differences in use and environmental features. Authors state that day-scale temporal trend were able to be detected, this seems like a fair statement due to the timing of when video data was taken.

Comment 1: The authors focus their conclusions around the enhanced ability these methods provide to efficiently analyse video data. These conclusions are in line with regards to the results, however, it would be of interest to hear how the results and the methods presented in this manuscript may be applicable to different systems. With the increased use of drones, video data is more frequently being collected for wild cetaceans, can CNN computational methods aid in analysing this data and what are the benefits or potential uses of using the methods presented in this manuscript.

Response: This is an excellent use case for CNN tracking, and is already at work in whale (Guirado, E. et al., “Whale counting in satellite and aerial images with deep learning”) and shark (Sharma N. et al., “Shark Detection from Aerial Imagery Using Region-Based CNN, a Study”) tracking in the wild. When transitioning to wild dolphin tracking, drones and gliders will likely be the vehicles of choice. The methods in this manuscript, particularly the tracklet generation, can be useful for not only identifying and locating the animals, but also in providing basic kinematics information on entire groups, which is not generally feasible with tagging operations. This has been noted in the last paragraph of “Limitations and future work” in the Discussion.

Page 16 Discussion, Limitations and future work – Paragraph 3: Further, the use of unmanned drones and gliders has the potential to extend this research for implementation in a wild setting. CNN tracking is already at work in whale [Guirado et al. 2019], and shark [Sharma et al. 2018] tracking in the wild, and the inclusion of these vehicles will open up new opportunities in making this research physically portable. The methods in this manuscript, particularly the tracklet generation, can be useful for not only identifying and localizing the animals, but also in providing basic kinematic information on entire groups, which is not generally feasible with tagging operations due to the limited number of tags available for deployment.

 - Guirado E, Tabik S, Rivas ML, Alcaraz-Segura D, Herrera F. Whale counting in satellite and aerial images with deep learning. Scientific Reports. 2019;9(1):1–12.

 - Sharma N, Scully-Power P, Blumenstein M. Shark detection from aerial imagery using region-based CNN, a study. In: Lecture Notes in Computer Science (including subseries Lecture Notes in Artificial Intelligence and Lecture Notes in Bioinformatics). vol. 11320 LNAI. Springer, Cham; 2018. p. 224–236.

Comment 2: Through describing a method that enables a more efficient analysis of video monitoring data, the authors have also created a greater understanding of the distribution (and potential environmental reasoning) for bottlenose dolphins present in this captive environment.

a. Are the difference observed in kinematics a natural occurring behaviour, a function of the environment, or due to different stimuli occurring at different times of day?

b. If the dolphins are spending the majority of their OTS at areas where more enrichment is occurring (windows, gates etc.) are there suggestions the authors could make about how these animals are managed?

Response: Answering these two questions thoroughly will involve additional experimentation regarding enrichment, which is the eventual goal. This has been noted in an addition made to “Limitations and future work” in the Discussion.

Page 16 Discussion, Limitations and future work – Paragraph 2: Beyond technical improvements, the next step in this research is to use long-term animal monitoring to inform management policy and aid welfare tracking. By working closely with the ACSs at the Brookfield Zoo, we intend to use the techniques presented in this manuscript to observe animal interactions with conspecifics and enrichment over time, track activity levels, and measure the effects of changing environmental conditions (e.g. effects of varying crowd size, spectator presence). In particular, given the emphasis the dolphins placed on particular regions of the environment, it will be important to evaluate the effects of attractors in these areas by varying enrichment type, physical placement, duration/time of exposure, and by recording ACS presence and interactions within these areas. In this way, we aim to guarantee a high level of animal engagement and work to identify potential stressors that may aid the Zoo in caring for their dolphins.

Comment 3: Expanding on the habitat use and kinematic results is relative because the title and points made throughout the manuscript suggest that habitat use and kinematics is the main focus and result. If this point is true than more emphasis should be place on discussing the implications of these findings for these dolphins. However, upon reading the manuscript, the impression is that the authors are describing new methods for analysing video data, and that habitat use and kinematics are potential uses for this tool. If this latter point is true, then the authors should consider changing the title to suggest a methods manuscript e.g. “Application of CNN in analysing video-based monitoring data for daily habitat use and kinematics of captive bottlenose dolphins”.

Response: We thank the reviewer for bringing this to light, and to ensure the title more closely matches the intent of the manuscript, it has been changed to “Computer-vision object tracking for monitoring bottlenose dolphin habitat use and kinematics”, which places the “object tracking” aspect in the forefront of the title.

Comment 4: A point of interest, the kinematics and habitat use of bottlenose dolphins was provided as a group summary, rather than for individuals. Do all dolphins consistently group together and follow the same path while in the enclosure, or, is there individual variation present. Can CNN analysis techniques detect and follow individuals through time? If the authors could provide comment on this I believe it would, at least, be a valuable discussion point as a future direction for this work.

Response: It is possible for CNN techniques to identify and track individuals in video, however, in this particular case the insufficient camera resolution and environmental occlusion effects (glare regions, water surface disturbances) make continuous tracking and individual identification impossible. This has been noted in an addition to “Detector and filter performance” in the Results section, and potential solutions to this problem are presented throughout the first paragraph of “Limitations and future work” in the Discussion section.

Page 11 Results, Detector and filter performance – Paragraph 3: A note on detector limitations and the animal identification problem: while this system is robust in detecting a dolphin in-frame, it cannot track specific animals. The camera resolutions are not sufficient to resolve identifying features on the animals, and the environmental occlusions (glare regions, severe water surface disturbances) prevent continuous tracking (Fig. 3, top). As a result, while each tracklet corresponds to a single dolphin at a time, the lack of identifiability prevents individual longer-duration tracking (>30 seconds) and therefore prevents individual metrics generation. For this reason, the results in this manuscript are presented for the dolphins as a group, rather than for each individual.

Comment 5: The authors have provided an ethics statement that lists the names of two organisations that approved this research. Do they have an ethics/permit number that could be referred to, that they could supply?

Response: Unfortunately, the ethics approval for research involving the dolphins at the Brookfield Zoo predates their IACUC system, and therefore does not have a listed reference number.

Specific Comments:

Comment 1: Page 1 Line 8: Authors state that “there is a strong emphasis on behavioral monitoring to inform welfare practices” but do not mention this again. Is there a suggestion that the continual video monitoring and use of CNN methods would be applicable to monitor behaviours linked to the welfare of these animals in the future? If so, the authors should provide greater detail into how these methods could benefit behavioural analyses.

Response: The authors’ intentions to expand upon this research to fully explore the welfare monitoring question has been included in the newly added second paragraph of “Limitations and future work”, in the Discussion.

Comment 2: Page 3 Line 75: Authors provide an average age of the seven bottlenose dolphins in this study (17 +/- 12). Due to the large range in ages I do not believe this metric is a good descriptor of dolphin ages and authors should simply provide the ages.

Response: The animals’ specific ages have been added to the text of the Materials and methods.

Page 3 Materials and methods, Experimental environment – Paragraph 1: Seven bottlenose dolphins of ages 5, 5, 14, 16, 17, 33, and 36 years with lengths of 247 +/- 17 cm were observed using a dual-camera system in the Seven Seas building of the Brookfield Zoo, Brookfield IL.

Comment 3: Page 8 Line 273 onwards: In this section the authors provided details on their heatmap production method. Supplying the software that was used in generating these heatmaps would be beneficial to readers and for reproducibility. Additionally, have these methods been described previously in other literature, or is this a method that was developed by the authors? Either way, authors should state how and why they used these methods when other commonly used methods exist (for example kernel density estimation, which can have a barrier function for use with hard limits to movement, such as the walls of a pool).

Response: The code used to perform the analysis will be provided as part of the final submission. The method for generating the heatmaps in this work was developed by the authors, and this specific kernel generation method was used as the volume of data, both in terms of animal positions in space collected by the cameras as well as the dolphins’ depth information collected prior via tagging, allowed for a direct generation of the heatmaps rather than needing to rely on estimation methods. This was noted in the “Position uncertainty” subsection of the Materials and methods.

Page 8 Materials and methods, Position uncertainty – Paragraph 1: Due to the high volume of data available to produce the underlying structure of the spatial distribution, the distribution kernels themselves could be directly generated rather than relying on estimation techniques.

Comment 4: Page 10 Line 325 onwards: Justification of the use of Kolmogorov-Smirnoff non parametric tests are warranted in this section. A statement of the assumptions and how the collected data has met these assumptions is important for assessing the appropriate use.

Response: The probability distributions describing the animals’ dynamics metrics did not conform to any standard set of distributions, and so a more generalized statistical comparison test was required. This has been indicated in the “Kolmogorof-Smirnov statistics” subsection of the Materials and methods.

Page 10 Materials and methods, Kolomogorov-Smirnov statistics – Paragraph 1: K-S statistics were chosen to allow for nonparametric comparisons between distributions, as the metric distributions within each subtype (e.g. animal speed, yaw) did not all pertain to the same family of distributions (e.g. normal, exponential, etc.), rendering more traditional statistical comparisons ill-suited to this application.

Comment 5: Page 14 Line 486 onwards: The linking of dolphin distribution to environmental features, such as gates and windows has sound logic, however, no formal quantitative tests were performed. The links made between these features and kinematics supports these assumptions but it is difficult to tell if this is a correlation or driver of habitat use. The authors could consider analysing the differences in habitat use using species distribution models.

Response: We thank the reviewer for pointing out this discrepancy, and the language has been softened in this section to reflect this. The inclusion of species distribution modeling is an interesting prospect, and while currently outside the scope of this paper, would be valuable to investigate in future work.

Page 15 Discussion, Habitat use – Paragraph 2: When combined with the results from the spatial distributions (Fig. 5, left), this implies that these dolphins not only focused their attention on these regions, their presence correlated to higher activity levels in the dolphins when swimming in their vicinity.

Reviewer #2

General Comments:

This study describes the implementation of a camera based auto-tracking approach to monitor dolphin locomotion in a managed area. The approach described is sound and the results suggest that it is effective for monitoring animal behavior, with the possibility to expand or enhance its performance through additional cameras or sensors. This work is consistent with other auto-tracking programs that have been described recently (reviewed in Panadeiro et al, 2021; https://doi.org/10.1038/s41684-021-00811-1), but is focused on the specific application of dolphins in an managed enclosure. Overall, I don't have any major concerns regarding this study. It is clear and well written, and will likely be of interest to scientists working in this area.

Specific Comments:

Comment 1: On line 101, formal training sessions are defined as ITS, but under Table 2, it states that OTS blocks 2 & 4 are formal presentations. What is the difference between formal training and formal presentations? Perhaps consider using more distinct language to describe these?

Response: We thank the reviewer for this suggestion, and in Tables 1 and 2 the term “formal presentations” has been changed to “public presentations” for clarity.

Reviewer #3

General Comments:

This article struggles between two possible narratives:

(1) a new ML-based method to track dolphins in captivity, with data to show the utility of the methods, and

(2) a dolphin distribution and behavior study in captivity that uses a ML-based method to compare behaviors while in two different behavioral conditions.

I believe that the goal of the authors is the first narrative and as such, my comments below are reflective of this.

Comment 1: Distribution is not the same as behavior and it feels like these are used interchangeably throughout. When you are talking about the position of the animal, you are referring to their distribution within the tank. As such, the word habitat use should also not be used as a tank is not a habitat. Space use in tank would be more appropriate. Behavior refers to the speed, yaw and dynamic swimming state.

Response: We thank the reviewer for pointing out this discrepancy, and to address this, use of the term “behavior” was limited to parts of the text that discuss animal dynamics states, and when discussing position was changed to “spatial distribution”. The contemporary term as used in zoo animal science for a zoo animal’s environment is “habitat”, and we have provided an abbreviated reference list of peer-reviewed papers that have used the term, both old and new:

 - Maple TL, Finlay TW. Applied primatology in the modern zoo. Zoo Biology. 1989;8(S1):101-116.

 - Powell DM, Baskir E. Behavior and Habitat Use Remain Diverse and Variable in Modern Zoological Exhibits over the Long-Term: Case Studies in 5 Species of Ursidae. Journal of Zoological and Botanical Gardens. 2021;2:677-704.

Comment 2: Statistics confirm that there are real, statistical differences and similarities in data. They aren't used to "quantify" them or "give clearer view". Be careful of how you use the KS statistics in the discussion. Really, they just tell you that the patterns, differences and similarities that you see are real and not due to lack of sample size.

Response: To correct this error, the language when discussing the K-S statistics has been modified to use more appropriate terms such as “confirm” or “identify”.

Page 13 Results, Statistical comparison of metrics for behavior differentiation – Paragraph 1: The K-S statistics were used to confirm the similarities and differences between time blocks within both OTS and ITS.

Page 16 Discussion, Behavior classification from dynamics metrics – Paragraph 2: In contrast, an analysis of the K-S results allowed for the identification of the statistical differences in animal dynamics between OTS time blocks.

Comment 3: Using the word "managed" is confusing as there are many wild populations that are managed. Use the word "captivity". Make it clear to the reader that this sort of study is only possible in captivity. This setup would not work in the wild.

Response: We thank the reviewer for identifying this potential point of confusion. To clarify that the animals are not wild dolphins, the use of the term “managed” has been replaced throughout the manuscript with the terms “zoo setting” and “under professional care”.

Specific Comments:

Comment 1: [Abstract] Lacks a "so what" or big picture. Why should someone want to read this article? What is the new innovation that helps to advance monitoring of dolphins in captivity?

Response: The Abstract has been re-worked to more clearly define the novelty of this research (“The resulting approach enables detailed persistent monitoring of the animals that is not possible using manual annotation methods.”) and its greater impact (“results from the proposed framework will enable future research that will offer new insights into dolphin behavior, biomechanics, and how environmental context affects movement and activity”).

Page 1 Abstract: The resulting approach enables detailed persistent monitoring of the animals that is not possible using manual annotation methods.

Page 1 Abstract: The work presented here demonstrates that CNN object detection is viable for large-scale marine mammal tracking, and results from the proposed framework will enable future research that will offer new insights into dolphin behavior, biomechanics, and how environmental context affects movement and activity.

Comment 2: This introduction does not prepare the reader for what is to come in the article and lacks reference to the body of work done with dolphins in captivity and in machine learning detection.

Response: We thank the reviewer for bringing this to light, and have restructured the Introduction to address this, taking into account the reviewers’ other comments on the section. Additional sources were included to support the choice of Faster R-CNN as the detector of choice for this research. Further, as existing research with captive animals is the focus of the second paragraph of the Introduction, additional text was added to highlight some of the foci of the research performed by direct animal monitoring within a captive environment.

Page 2, Introduction – Paragraph 2: Examples of such studies include monitoring the effects of human presence on animal behaviors, analysis of dolphin activity cycles and sleep patterns, and the evaluation of social interactions with conspecifics.

Page 2, Introduction – Paragraph 3: Faster R-CNN [Ren et al. 2017], was chosen as the backbone of the animal detection method for its prioritization of accuracy and precision regardless of object size or density in the image, as opposed to a faster single-shot detector [Redmon and Farhadi 2017]. The Faster R-CNN detector structure has demonstrated its capabilities in both land [Manning et al. 2019] and marine [Hsu et al. 2021] applications, and is considered a reliable option for challenging tracking tasks.

 - Ren S, He K, Girshick R, Sun J. Faster R-CNN: Towards Real-Time Object Detection with Region Proposal Networks. IEEE Transactions on Pattern Analysis and Machine Intelligence. 2017;39(6):1137–1149.

 - Redmon J, Farhadi A. YOLO9000: Better, Faster, Stronger. In: Proceedings of the IEEE Conference on Computer Vision and Pattern Recognition (CVPR);2017.

 - Manning T, Somarriba M, Roehe R, Turner S, Wang H, Zheng H, et al. Automated Object Tracking for Animal Behaviour Studies. In: Proceedings - 2019 IEEE International Conference on Bioinformatics and Biomedicine, BIBM2019. Institute of Electrical and Electronics Engineers Inc.; 2019. p. 1876–1883.

 - Hsu HM, Xie Z, Hwang JN, Berdahl A. Robust fish enumeration by multiple object tracking in overhead videos. In: Lecture Notes in Computer Science (including subseries Lecture Notes in Artificial Intelligence and Lecture Notes in Bioinformatics). vol. 12662 LNCS. Springer, Cham; 2021. p. 434–442.

Comment 3: The statement "Here we present an automated..." in paragraph 1 comes too soon. You haven't even described any of the previous research in the field. This should be in your last paragraph.

Response: The statement has been removed.

Comment 4: The paragraph about tags in the wild "Biomechanics and behavioral studies..." does not add anything to the article and is not relevant given that the entire study is done in captivity and there is no comparison or attempt to implement this in the wild.

Response: The components of the paragraph discussing tag usage and wild animal studies have been removed, and the remainder was merged into the following paragraph.

Comment 5: I did not feel like the last paragraph that describes this study is accurate. It did not prepare me for what was to come. Please make sure they match in content and breadth as well as order ideas are presented in methods.

Response: The final paragraph of the introduction has been replaced with a modified version of the first paragraph of Materials and methods, which more directly states the methods that are used and the intent with which they were applied.

Page 2 Introduction – Paragraph 4: In this study, camera data were used to monitor the behavior of a group of marine mammals both qualitatively and quantitatively in a zoo setting. Camera-based animal position data were used to quantify habitat usage, as well as where and how the group of animals moved throughout the day. The position data were decomposed into kinematic metrics, and used to discriminate between two general movement states - static and dynamic - using the velocity of the tracked animals. A general ethogram of the animals' behaviors monitored in this research is presented in Table 1. Joint differential entropy computations were calculated using animal speed and heading data to provide an understanding of the dolphins' kinematic diversity. Kolmogorov-Smirnov statistical analyses of the kinematic metrics were used to compare movement patterns and activity levels over time and between behavioral conditions. The proposed framework and results presented here demonstrate the viability of computer-vision inspired techniques for this challenging monitoring problem, and will enable future studies to gain new insights into dolphin behavior and biomechanics.

Comment 6: Suggested structure: paragraph 1 = what we've previous learned from behavior studies of captive animals, paragraph 2 = methods used to study captive animals and limitations, paragraph 3 = overview of machine learning detection algorithms showing other applications of RPN and Fast R-CNN (since you are not inventing them in this study) and remaining gaps, paragraph 4 = here's what we are going to do and show in this article.

Response: The suggested structure has been taken into consideration, with the modification of more of a focus on how monitoring has been performed, rather than what has been learned, as this manuscript does focus more on the tracking method than an extensive behavior review. A note has also been made in the second paragraph to provide examples of what has been performed prior to this work.

Page 2 Introduction – Paragraph 2: Examples of such studies include monitoring the effects of human presence on animal behaviors, analysis of dolphin activity cycles and sleep patterns, and the evaluation of social interactions with conspecifics.

Comment 7: Overall, methods are not organized in a manner that the reader feels like they build on each other. We switch from behavior related things to ML related things. Too many details on the behavior stuff given that its not the focus.

Response: We have removed the excess detail on dolphin behavior reinforcement. However, the remainder provides a basic explanation of the ITS time block structure which aids in interpreting the ITS spatial maps, and these portions were left in for this reason.

Comment 8: First paragraph in M&M feels unnecessary and almost more appropriate to do in intro.

Response: We thank the reviewer for this suggestion, and have moved this paragraph to the end of the Introduction and modified it for flow.

Comment 9: None of the details about the tank feel necessary nor relevant as they are really never revisited. Maybe add these measurements to the top panel of figure 1 and then there is no need to repeat them in text.

Response: The unnecessary information on secondary habitats has been removed. Unfortunately, the structure of Figure 1 makes placement of the main habitat dimensions somewhat awkward, so these are left in the text.

Comment 10: Figure 1 and 2 are way too busy and need to be broken into different panels. I would combine top panel of fig1 and fig 2 as 1 figure. Tracking algorithm its own figure and prob dist their own as well. Label panels (a), (b), (c) as top, bottom etc. didn't really help because there were so many panels in each figure.

Response: The reviewer’s suggestions on the figures’ structures helped immensely in their simplification. Addressing these suggestions, the top panel of Figure 1 and a simplified top panel of Figure 2 were combined into a single figure for the environmental setup. The bottom panel (including popouts) of Figure 1 was isolated as a separate tracklet figure. Additionally, an image demonstrating tracklet examples was added to the tracklet figure. Figure 2 had unnecessary components of the top panel removed, but the panel was retained to support the meshing computation explanation.

Page 3 Materials and methods – Figure 1:

Fig. 1. Diagram of the experimental setup. TOP: Illustration of the main habitat, with camera placements (blue enclosures) and fields of view (gray cones). BOTTOM: Top-down individual camera views, with objects in the habitat marked. Yellow -- Dolphin bounding boxes, Green -- Drains, Red -- Gates between regions, Orange -- Underwater windows (3 total). Correlated dolphin bounding boxes are indicated by number.

Page 6 Materials and methods – Figure 2:

Fig. 2. Combined figure demonstrating camera overlap, bounding box meshing, and animal position uncertainty. TOP: Top-down individual camera views, with dolphin bounding boxes in yellow (correlating boxes are numbered). The habitat-bisecting lines (l_s) for each camera frame are indicated in solid red. Distances from Bounding Box 2 (centered on the black and gray crosshair) to the closest frame boundary (d_b) and the boundary to the bisecting line (d_l) are indicated by the white measurement bars. MIDDLE: Meshed camera views including dolphin bounding boxes (yellow), with the location uncertainty distribution (A) overlaid for Box 2. BOTTOM: 2D location uncertainty distribution (A) with major (a-a, black) and minor (b-b, red) axes labeled and separately plotted.

Page 7 Materials and methods – Figure 3:

Fig. 3. Illustration of tracklet generation. TOP: Tracklet segments (red) overlaid on a single video frame, generated by stitching the views from both cameras. Each tracklet in this frame was plotted from its inception to each corresponding dolphin's current position. While each dolphin can be tracked, the lack of clarity when underwater impedes individual identification. CENTER: x-y view of example tracklets (red and green on gray lines) of two dolphins (highlighted light orange), which are also shown in Fig. 1, top. POPOUT-RIGHT: Vector illustrations of the two example tracks. Example notation for tracklet j (red): position (p^(j,t')), velocity (v^(j,t')), yaw (θ^(j,t')), and yaw rate (θ ˙^(j,t')). POPOUT-BOTTOM Illustration of tracklet generation, with detections (stars) and tracklet proximity regions (dashed). Example notation for tracklet j (red): position (p^(j,t)), velocity (v^(j,t)), Kalman-predicted future position ( p ^^(j,t+1)), true future position (p^(j,t+1)), and future animal detection (u^(j,t+1,i')).

Comment 11: Table 2. Add your speed, yaw metrics in this table with the time intervals.

Response: The speed and yaw rate metrics have been added to Table 2.

Page 4 Materials and methods – Table 2:

Table 2. The ITS blocks (1 and 3) are animal care sessions, and the OTS blocks (2 and 4) are public presentations. The corresponding mean speed and yaw rate dynamics metrics are also reported, with yaw rate converted to units of (deg s^-1) for readability.

Comment 12: What are the metrics of detector performance?

Response: The Faster R-CNN detectors were fully evaluated in Zhang D et al., “Localization and tracking of uncontrollable underwater agents: Particle filter based fusion of on-body IMUs and stationary cameras.” A reference to this paper was included in “Detector and filter performance” in the Results section, along with a summary of the results and a table providing the performance metrics of the tracking system with respect to ground-truth. The summary text and table are provided in the responses to Comments 19 and 20, respectively.

Comment 13: False positives should be their own section.

Response: False positives moved to a separate subsection, “False positive mitigation”.

Comment 14: Tracklet is first used in title so doesn't lend well to preparing reader. Use this term in intro when describing what is going to be done.

Response: The subsection title has been changed to “Temporal association of detections”.

Comment 15: Should reference tracklet figure earlier on in this section.

Response: The tracklet figure is now referenced in the first paragraph of the “Temporal association of detections” subsection in the Materials and methods.

Comment 16: What is proximity region? How was it defined?

Response: The proximity region definition has been added in the second-to-last paragraph of “Temporal association of detections” in Materials and methods.

Page 8 Materials and methods, Temporal association of detections – Paragraph 3: Using the predicted position, the k-th tracklet checked whether there existed a closest detection in the next frame that was within the proximity region of the predicted position, which is defined as a circle around the predicted position with radius 0.8 m (heuristically tuned).

Comment 17: Heatmap representation is really confusing given the aims of the study to track. Why isn't an individual track shown? This would be more powerful representation that the left column panels in figure 4 and 5.

Response: Due to individual identification and environmental occlusion problems (seen in Fig. 3, top, where the tracklets will cut out when a dolphin passes below an occlusion region), a single tracklet for one animal only has a duration on the order of tens of seconds before it is deactivated due to inactivity, and the animal is then picked up later by another tracklet. It would be preferable to have individual tracks instead of heatmaps to show, but unfortunately the system cannot accomplish this as-is (this is the focus of our ICRA paper: Zhang D et al., “Localization and tracking of uncontrollable underwater agents: Particle filter based fusion of on-body IMUs and stationary cameras.”) An additional paragraph has been added to address this in “Detector and filter performance” in the Results section.

Page 11 Results, Detector and filter performance – Paragraph 3: A note on detector limitations and the animal identification problem: while this system is robust in detecting a dolphin in-frame, it cannot track specific animals. The camera resolutions are not sufficient to resolve identifying features on the animals, and the environmental occlusions (glare regions, severe water surface disturbances) prevent continuous tracking (Fig. 3, top). As a result, while each tracklet corresponds to a single dolphin at a time, the lack of identifiability prevents individual longer-duration tracking (>30 seconds) and therefore prevents individual metrics generation. For this reason, the results in this manuscript are presented for the dolphins as a group, rather than for each individual.

Comment 18: Suggested structure: section 1 = camera setup in tank area + how much data was collected over what periods of time, section 2 = manual labelling and analysis, section 3 = neural net description + training + metrics of detect performance, section 4 = detection processing to combine frames, 5 = position uncertainty, 6 = tracklets + heatmaps to visualize, 7 = drain detector, 8 = about different behavior states and dolphins training sessions

Response: We thank the reviewer for the structure suggestion, however, the structure of the Materials and Methods was kept primarily as-is to more easily cluster the types of sections together. Physical equipment, environment, and test subjects were stated first, to indicate what we are working with and where, and behavior states were indicated here to prepare the reader for what we were intending to monitor. The neural network methods were then all stated in order of computation (however in this case the Training subsection was stated after defining the network to provide an explanation of what was being trained, in case readers are not familiar with CNN-based object detection). Tracklet computation was stated immediately after to provide information on how both location filtering and dynamics extraction functioned. Spatial post-processing methods (position uncertainty and heatmap generation) and dynamics post-processing methods (joint entropy and K-S statistics) could then be ordered with either one first, however, as the focus of the section had so far remained on localization, it followed to place spatial use first and dynamics second.

Comment 19: Given that this article seems to focus a lot on the detector and tracklet, there is only a single paragraph describing the performance of these algorithms and most of it is about the detector. How was the tracklet algorithm performance? What there any human ground-truthing of animal trajectory to confirm the tracklet trajectories?

Response: The detectors used in this research were also employed in a separate study: Zhang D et al., “Localization and tracking of uncontrollable underwater agents: Particle filter based fusion of on-body IMUs and stationary cameras.” Their performance was fully evaluated in that paper, and these results have been summarized in the first paragraph of “Detector and filter performance” in the Results, along with the addition of a table providing the specific values (the table is located in the response to the immediately following comment).

Page 10 Results, Detector and filter performance – Paragraph 1: The performance of this pair of Faster R-CNN detectors with comparisons to ground truth was fully evaluated in [Zhang et al. 2019], with the results reported in Table 3. To summarize the performance results: two additional monitoring sessions were video recorded and tracked both manually and using the automated CNN-based tracking system from this manuscript. During these sessions, two individual dolphins were tracked by a human observer and the results were compared to the detections produced by the automated system. Overall, for these two deployments the human tracker (representing the ground-truth) was able to detect the dolphins 88.1% of the time, while the CNN-based trackers were able to detect the dolphins 53.2% of the time. As a result, the automated system achieved an overall recall rate of 60.4% versus ground-truth.

 - Zhang D, Gabaldon J, Lauderdale L, Johnson-Roberson M, Miller LJ, Barton K,et al. Localization and tracking of uncontrollable underwater agents: Particlefilter based fusion of on-body IMUs and stationary cameras. In: Proceedings -IEEE International Conference on Robotics and Automation. vol. 2019-May.Institute of Electrical and Electronics Engineers Inc.; 2019. p. 6575–6581.

Comment 20: Make a table to summarize performance of algorithms.

Response: The performance table has been provided (Table 3).

Page 11 Results – Table 3:

Table 3: Performance comparison between manual and CNN animal detections for two sessions as part of a separate monitoring exercise, where individual dolphins were tracked as opposed to the entire group. A1 and A2 refer to specific dolphins, with A1 being tracked over two recordings during Deployment 1, and A1 and A2 being tracked during the same recording during Deployment 2. “Detectability” is defined as the total time each individual dolphin was able to be detected by either the human or CNN trackers over each deployment period.

Comment 21: The comparison between OTS and ITS feels odd given that it is framed as comparing behavioral states and locations of the animal. In ITS, it is really obvious that the dolphins are going to be where the algorithm finds them in front of the stage area because that is where the trainer tells them to be. So it really isn't a "distribution" or their space use in the tank. I would frame this more as a confirmation of your detector working because the detector found the dolphins where they are supposed to be and maybe should be included in detector performance.

Response: In this manuscript, the primary comparison of OTS versus ITS in terms of animal behavior is framed through the analysis of the dynamics metrics. Conversely, animal spatial use is treated as a consequence of their behavior modes, and comparisons are made to highlight how spatial use changes with regard to behavior, if at all. In general, confirmation and quantification of spatial distributions of animals based on behavior mode represents a contribution unto itself, and this work demonstrates a simple representation of position dependent on behavior. It is true that observing the dolphins in regions where they are expected to be does provide validation of the performance of the detection framework, although this would primarily be qualitative rather than quantitative, which is noted in “Spatial distribution – position” in the Results.

Page 12 Results, Spatial distribution – position – Paragraph 2: This also serves to qualitatively validate that the detectors are working as expected, given the dolphins are observed to be present in a region they are commonly instructed to occupy.

Comment 22: Figure 4 and 5. The differences in the space use don't really stand out with these plots. It may be best to plot a single track of an animal in OTS and one in ITS to show the space use. There's also a ton of text in the figure captions that isn't mentioned in the main text.

Response: Unfortunately, the current state of the detection framework does not allow for individual animal identification, and as such when the animals enter and exit occlusion regions, or exit the cameras’ views entirely, re-acquisition of any particular individual is not possible. Effectively, each animal can be tracked, but any occlusion leads to loss of identity without manual intervention. An additional image has been added to the top of the tracklet figure (now Fig. 3) to illustrate this problem. An example video showing the tracking performance has been prepared for the supplementary materials, which will further support this statement. We thank the reviewer for pointing out the discrepancy with the figure captions, of which the ITS figure caption particularly needed editing (previously Fig. 5). The explanatory text from Fig. 5 (now Fig. 6) has been moved to the “Spatial distribution – speed/quiver” subsection of the Results.

Page 13 Results, Spatial distribution – speed/quiver – Paragraph 2: In general, speeds across the entire habitat are higher during public presentations than non-public animal care sessions because high-energy behaviors (e.g., speed swims, porpoising, breaches) are typically requested from the dolphins several times throughout the presentation. Though non-public presentations include high-energy behaviors, non-public animal care sessions also focus on training new behaviors and engaging in husbandry behaviors. Public presentations provide the opportunity for exercise through a variety of higher energy behaviors, and non-public sessions afford the ability to engage in comprehensive animal care and time to work on new behaviors.

Page 12 Results – Figure 6 (caption): Spatial distributions for dynamic ITS, with position distributions along the first column and speed distributions/quiver plots along the second column. During the animal care sessions (Block 1: 09:30 to 10:00, Block 3: 13:00-13:30), the dolphins engaged in lower intensity swimming throughout the habitat than the presentation sessions (Block 2: 11:30-12:00, Block 4: 14:30-15:00). This difference is qualitatively explained through the discrepancy in ACS requests from the animals: high-intensity behaviors are prompted more often during presentations, while care sessions cover a wider variety of behaviors. Conversely, spatial coverage of the habitat does not have high variance within the ITS blocks, with an expectedly high concentration on the central island where the ACSs are located for all ITS blocks.

Comment 23: Table 3 should be in the supplementary.

Response: Table 3 has been moved to the supplementary section as Table S1.

Comment 24: I don't think yaw is ever actually defined or described.

Response: We thank the reviewer for catching this error, and a clarification on the use of yaw in this paper has been included in the “Dolphin detection” subsection in Materials and methods.

Page 5 Materials and methods, Dolphin detection – Paragraph 1: Kinematic data in the form of position, velocity, and yaw (heading in the x-y plane), from the tracklets were then used to parameterize and form probability distributions …

Comment 25: Statistical comparison and entropy should not be their own sections but rather woven into the Speed + Yaw description of the animals.

Response: For this manuscript, the statistical comparisons were intended specifically to evaluate if metric distributions could differentiate overall group behavioral states. Alternatively, the joint entropy results were intended to evaluate the group’s kinematic diversity. As such, the titles have been changed to reflect these particular foci: “Joint entropy results for kinematic diversity analysis” and “Statistical comparison of metrics for behavior differentiation”.

Comment 26: Suggested structure: section 1 = detector performance, 2 = tracklet performance, 3 = animal space use during OTS and ITS, 4 = speed + yaw of animals

Response: We thank the reviewer for the structure suggestion for the Results section. However, the structure was kept as-is as the tracklet performance is heavily linked to the overall detector framework performance, and keeping these together is more conducive to the flow of the text. The Results section was otherwise restructured according to the reviewers’ additional comments on the Discussion section, as some components of the Discussion were more appropriate for placement within the Results.

Comment 27: Most of the paragraphs in the discussion read as results paragraphs and should be moved to the results. These paragraphs explain the results so much more clearly than the results section. I really didn't comprehend the results of the article until I read the discussion section.

Response: We thank the reviewer for this suggestion, and to address this the components of the Discussion section that pertained more closely to Results have been moved. The component of the first paragraph of “Kinematic diversity” that pertained to the Results has been moved to “Joint entropy results for kinematic diversity analysis” in the Results (text found in the response to the immediately following comment). The components pertaining to the Results in “Behavior classification from dynamics metrics” have been merged into “Statistical comparison of metrics for behavior differentiation” in the Results.

Pages 13-14 Results, Statistical comparison of metrics for behavior differentiation – Paragraphs 1-2: The K-S statistics were used to confirm the similarities and differences between time blocks within both OTS and ITS. To aid in visualizing this, Figure 7, top, displays the overlaid PDFs of the speed and yaw metrics during OTS, and Figure 7, middle, displays the PDFs during ITS. A complete table with K-S distances and α values for all six metrics is present in Table S1 in the supplementary materials, with all values rounded to 3 digits of precision. For OTS, we saw from the K-S results that Blocks 1 and 2 varied the most with respect to the others in terms of speed, which was observed in Figure 7, top, while the yaw values were not generally significantly different, again observed in Figure 7 (given the high number of samples used to generate the K-S statistics, we were able to compare the significance levels to a stronger threshold of α_crit=0.001). Across the board, Block 2 generally differed significantly from the rest of the OTS blocks for the most metrics, with Block 1 following close behind. In contrast, Blocks 3-5 differed the least significantly from each other, indicating similarities in the dolphins' dynamics patterns for Blocks 3-5.

For ITS, we note that the significant differences in metrics generally followed the structure type of each ITS block: comparisons between Blocks 1 vs. 3, and 2 vs. 4, were found to be significantly different the least often. As the ACSs requested similar behaviors during ITS blocks of the same type, we expected similarities in the dynamics metrics for Blocks 1 vs. 3 (animal care sessions) and Blocks 2 vs. 4 (presentations), and differences between the metrics for blocks of different types. Of particular note are the yaw std. dev. and yaw rate std. dev. metrics, with entire order of magnitude differences in K-S distances when comparing similar vs. different types of ITS blocks. Overall, the pattern displayed by the ITS K-S statistics in Table S1 correlated with this expectation.

Comment 28: section kinematic diversity = 1st paragraph reads like a results paragraph

Response: The results components of this paragraph have been moved into the “Joint entropy results for kinematic diversity analysis” subsection of the Results, and the “Kinematic diversity” subsection of the Discussion has been modified to reflect this change.

Page 13 Results, Joint entropy results for kinematic diversity analysis – Paragraph 1: The time blocks in this figure are presented in chronological order, and we observed the lowest kinematic diversity in the mornings (the first blocks of each OTS and ITS) as the animal care specialists were arriving at work and setting up for the day. The highest kinematic diversity when not interacting with the ACSs then occurred immediately after the first ITS time block. In general, the first time blocks of both OTS and ITS showed the lowest kinematic diversity of their type, the second of each showed the highest, and the following blocks stabilized between the two extremes. The speed/quiver plots (Figs. 5-6, right) provide a qualitative understanding of the entropy results. For example, in Block 1 of OTS (Fig. 5, top-right) the dolphins engaged in slow swimming throughout their habitat in smooth consistent cycles along the environment edge, yielding the lowest joint entropy. Joint entropy then increased during both the morning ITS and OTS blocks and remained elevated for the rest of the day, representing higher animal engagement through the middle of their waking hours.

Pages 14-15 Discussion, Kinematic diversity – Paragraph 1: The kinematic diversity results presented here are consistent with previous research on animal activity and sleep patterns, which reports a diurnal activity cycle for animals under professional care [Sekiguchi et al. 2003].

Comment 29: section behavioral classification = where these results even mentioned in the results section? it feels very unclear

Response: In line with the reviewer’s additional comments, the Results and Discussion behavior classification subsections have been restructured to better present the data in the former and their analysis in the latter.

Comment 30: Order the kinematic diversity, habitat use and behavioral classification in the same order as the results. (i.e., habitat use, behavioral classification and kinematic diversity??)

Response: We thank the reviewer for this suggestion, and have resolved the ordering problem.

---

## [Editor Report · Decision Letter 1]

17 Jan 2022

Computer-vision object tracking for monitoring bottlenose dolphin habitat use and kinematics

PONE-D-21-19199R1

Dear Dr. Gabaldon,

We’re pleased to inform you that your manuscript has been judged scientifically suitable for publication and will be formally accepted for publication once it meets all outstanding technical requirements.

Kind regards,

William David Halliday, Ph.D.

Academic Editor

PLOS ONE
---

## [Editor Report · Acceptance letter]

25 Jan 2022

PONE-D-21-19199R1 

Computer-vision object tracking for monitoring bottlenose dolphin habitat use and kinematics 

Dear Dr. Gabaldon:

I'm pleased to inform you that your manuscript has been deemed suitable for publication in PLOS ONE. Congratulations! Your manuscript is now with our production department. 

Kind regards, 

on behalf of

Dr. William David Halliday 

Academic Editor

PLOS ONE